# AnoLLM: Large Language Models for Tabular Anomaly Detection

**Che-Ping Tsai**,[*] **Ganyu Teng, Phil Wallis, Wei Ding**
Amazon
{chepingt, tenganyu, phwallis, dingwe}@amazon.com

## Abstract

We introduce AnoLLM, a novel framework that leverages large language models (LLMs) for unsupervised tabular anomaly detection. By converting tabular data into a standardized text format, we further adapt a pre-trained LLM with this serialized data, and assign anomaly scores based on the negative log likelihood generated by the LLM. Unlike traditional methods that can require extensive feature engineering, and often lose textual information during data processing, AnoLLM preserves data integrity and streamlines the preprocessing required for tabular anomaly detection. This approach can effectively handle mixed-type data, especially those containing textual features. Our empirical results indicate that AnoLLM delivers the best performance on six benchmark datasets with mixed feature types. Additionally, across 30 datasets from the ODDS library, which are predominantly numerical, AnoLLM performs on par with top performing baselines.

## 1 Introduction

Anomaly detection (AD) seeks to examine specific data points to identify rare, or specious occurrences that deviate from established behavior patterns. AD has been applied to a wide range of applications spanning computer vision (Chandola et al., 2009), natural language processing (NLP) (Schölkopf et al., 2001) and tabular data (Hawkins, 1980). Among these, tabular anomaly detection is particularly crucial, as tabular data is a fundamental format in machine learning which has been used extensively in applications related to cyber-attack prevention (Landauer et al., 2023), detecting fraudulent financial transactions (Dornadula & Geetha, 2019), and diagnosing medical conditions (Fernando et al., 2021).

Conversely, as large language models (LLMs) have shown remarkable performance on various NLP tasks (Radford et al., 2018; Chung et al., 2024; Wei et al., 2022), researchers are keen to understand the capabilities of LLMs applied to other modalities, including tabular data (Fang et al., 2024). While previous studies have demonstrated the proficiency of LLMs in prediction (Dinh et al., 2022), table understanding (Sui et al., 2024), and data generation (Borisov et al., 2023), their effectiveness in handling the task of tabular anomaly detection remains largely unexplored.

Applying LLMs to tabular anomaly detection presents several challenges. First, tabular data is inherently structured in two dimensions, which does not align well with the linear and sequential nature of LLM inputs. Second, unlike traditional tabular classification tasks, unsupervised anomaly detection lacks labels, making the in-context learning framework unfeasible. Third, while existing studies demonstrate that LLMs can be fine-tuned as generative models for tabular data, naively using output probabilities as anomaly scores introduces length-bias issues as the token-level probabilities are aggregated.

In this paper we introduce AnoLLM, a novel framework of using LLMs for unsupervised tabular anomaly detection. AnoLLM is comprised of three phases. In the initial phase, we serialize each row of a tabular dataset into a standardized text format. During the training phase, a pretrained LLM is fine-tuned with the before mentioned serialized tabular data via next-token-prediction, where the LLM learns to be a sequential tabular data generator that models the underlying data distribution.

---

[*]Work completed during an internship at Amazon

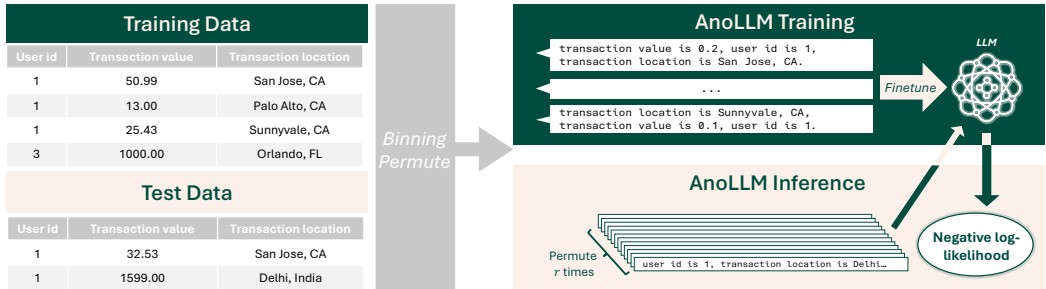

Figure 1: Overall framework of AnoLLM. During the preprocessing stage, numerical columns are binned into groups, and each data row is transformed into a natural language sequence with a randomly shuffled order of columns. In the training stage, a pretrained LLM is fine-tuned using the preprocessed tabular data. During inference, anomaly scores are determined by averaging the negative log-likelihood across $r$ random permutations of the test data.

In the inference phase, anomaly scores are determined using the negative log likelihood produced by the LLM for the test samples provided, with higher scores indicating greater surprise by the model when encountering the inputs. Since naively multiplying all token probabilities is prone to length-bias issue, we propose to normalize probabilities for each textual column individually, while retaining the raw probabilities for numerical and categorical columns, as supported by theoretical validation.

AnoLLM has the following advantages over existing tabular anomaly detection approaches. First, existing approaches often discard feature names and categorical values when transforming data into vectors, leading to a potential loss of valuable information. Second, these methods rely heavily on feature engineering to handle mixed-type tabular data (Borisov et al., 2022), such as text data, which cannot be easily transformed into numerical and categorical features. For instance, system log anomaly detection requires extensive feature engineering to transform logs into real-valued vectors (He et al., 2017). Moreover, existing approaches often require the use of imputers to handle missing values effectively (Emmanuel et al., 2021). In contrast, using text representations as inputs offers a more flexible and comprehensive representation of data compared to traditional anomaly detection models.

Our contributions are summarized below.

- We propose AnoLLM, a novel framework for adapting LLMs for unsupervised tabular anomaly detection. To the best of our knowledge, AnoLLM is the first tabular AD approach that is capable of handling raw textual features without pre-processing.

- To mitigate the length-bias in the LLM's output probabilities, we propose a novel method for computing anomaly scores based on LLM-generated probabilities, backed by theoretical validation.

- Empirically, we demonstrate that AnoLLM outperforms all existing methods, including four classical approaches and eight deep learning-based approaches, across six datasets containing mixed tabular data types.

- Despite known limitations of LLMs in performing basic arithmetic calculations (Shen et al., 2023; Lee et al., 2023a), our empirical evidence shows that the overall performance of AnoLLM matches best-performing baselines, k-nearest neighbors, internal contrastive learning, and diffusion time estimation, on 30 datasets in the ODDS library (Rayana, 2016), which primarily contain numerical features.

We also conduct extensive analysis on the key design choices in AnoLLM, including various feature binning methods and the impact of different LLM sizes. To the best of our knowledge, we are the first to apply LLMs to tabular anomaly detection, achieving state-of-the-art results.

## 2 ANOLLM: LARGE LANGUAGE MODELS FOR MIXED-TYPE TABULAR ANOMALY DETECTION

This section introduces the AnoLLM approach for detecting anomalies in tabular data, which comprises three main stages. An overview of the AnoLLM approach is shown in Figure 1. In the first stage, tabular data is serialized into text format. During the training stage, the serialized data is used to fine-tune pretrained large language models through next token prediction. In the inference stage, anomaly scores are assigned based on the negative log-likelihood generated by the LLMs for the given test samples. Each stage is described in detail below.

### 2.1 PROBLEM FORMULATION

We focus on uncontaminated, unsupervised anomaly detection using tabular data, represented as $\mathbf{X} = \{\mathbf{x}_1, \cdots, \mathbf{x}_n\}$, where $\mathbf{x}_i \in \mathcal{X}$ is the $i^{th}$ row of the data. The objective is to train an anomaly detector, $f : \mathcal{X} \mapsto \mathbb{R}$, solely on the normal training samples $\mathbf{X}$. The anomaly detector assigns scores to each sample as $f(\mathbf{x})$, with higher scores indicating a higher likelihood of being an anomaly.

During the test stage, we are provided with a labeled test set, $\mathbf{D}' = \{(\mathbf{x}'_1, y'_1), \cdots, (\mathbf{x}'_n, y'_n)\}$, where each $\mathbf{x}'_i \in \mathcal{X}$ and $y'_i \in \{0, 1\}$ with anomalies being labeled as $y = 1$. After applying the anomaly detector $f$ to each test sample, we obtain anomaly scores $\{f(\mathbf{x}'_1), \cdots, f(\mathbf{x}'_m)\}$. A threshold, or probability region, can be employed to classify whether a sample is an anomaly based on these scores.

Suppose that we have a tabular dataset with $n$ rows and $d$ columns (features). We denote the column (feature) names as $\mathbf{c} = [c_1, \cdots, c_d]$, with each $c_i$ being a natural language text sequence such as "age" or "job description". Let the value of each entry be $\mathbf{x}_{i,j}$ with $i \in \{1, \cdots, n\}$ and $j \in \{1, \cdots, d\}$. We note that $\mathbf{x}_{i,j}$ can represent different types of features, such as numerical, categorical or textual features.

### 2.2 SERIALIZATION OF TABULAR DATA

For serialization, we follow previous tabular data serialization methods (Hegselmann et al., 2023; Borisov et al., 2023) by simply concatenating column names and column values. Specifically, we use the following template: *column name* is *column value*. Formally, the $j^{th}$ column of the $i^{th}$ sample $\mathbf{x}_i$ is transformed into the text sequence $t_j(\mathbf{x}_i) = "c_j$ is $E(\mathbf{x}_{i,j})"$, where $E(\cdot)$ is a text encoder that pre-process the feature into the text form, which is introduced below.

**Feature preprocessing:** Since LLMs only take text as inputs, we use a text encoder $E(\cdot)$ to transform every row of data into text format. Specifically, for textual columns and categorical columns, we directly use their original form as they are already represented in natural language. For numerical columns, we do standard rescaling and then round them to single-digit decimals. This operation effectively reduces the number of digits when the inputs are very large, or have high precision. Formally, we apply the following affine transformation to each element $\mathbf{x}_{i,j}$.

$$E(x_{i,j}) = \frac{1}{10} \left\lfloor 10 \times \frac{\mathbf{x}_{i,j} - m_j}{z_j} \right\rceil, \text{ if } c_j \text{ is a numerical column.} \tag{1}$$

Here, $m_j$ and $z_j$ are the mean and the standard deviation of the $j^{th}$ column in the training set, i.e. $m_j = \text{mean}(\mathbf{x}_{1,j}, \mathbf{x}_{2,j}, \cdots, \mathbf{x}_{n,j})$, and $z_j = \text{std}(\mathbf{x}_{1,j}, \mathbf{x}_{2,j}, \cdots, \mathbf{x}_{n,j})$. $\lfloor \cdot \rceil$ is a rounding-to-integer operator. This affine transformation normalizes the numerical values and converts them into a decimal number with a single digital place. This operation effectively converts large or long-digit numbers into decimal numbers containing only a few digits. This step is crucial because we obtain the anomaly scores by computing negative log-likelihood, which is sensitive to token lengths.

This standard rescaling operation can be viewed as a feature binning method since most of the numerical values fall into a range centered around the mean $0.0$, containing a finite number of single-digit decimal numbers. In this approach, we categorize numerical values and allow AnoLLM to treat these numerical attributes as categorical features. The categories are represented by single-digit decimals that retain information related to magnitude. We choose to use standard rescaling for

our binning method since it outperformed other feature binning methods during experimentation, such as equal-width binning and percentile binning, as shown in Section 3.3.

**Handling missing values and column names:** For all missing feature values, the text encoder $E$ treats them as a separate category and maps them to the word *"Unknown."* For tabular data with missing column names, we manually assign names in alphabetical order as follows: *"A", "B", ..., "Z", "AA", "AB", ..., "AZ", "BA", "BB"*,, etc. As we will demonstrate in the experiments, AnoLLM remains effective even if the column names are not provided.

**Random column permutations:** Since there is no inherent ordering among columns, we permute the $d$ serialized text sequences corresponding to each column using a randomly selected permutation $\pi \sim S_d$, where $S_d$ is a symmetric group of $d$ elements, containing all $d!$ possible permutations. The resulting text sequence representing the row $\mathbf{x}_i$ is as follows:

$$\mathbf{T_c}(\pi, \mathbf{x}_i) = \text{``} \underbrace{c_{\pi(1)} \text{ is } E(\mathbf{x}_{i,\pi(1)})}_{t_{\pi(1)}(\mathbf{x}_i)}, \underbrace{c_{\pi(2)} \text{ is } E(\mathbf{x}_{i,\pi(2)})}_{t_{\pi(2)}(\mathbf{x}_i)}, \cdots, \underbrace{c_{\pi(d)} \text{ is } E(\mathbf{x}_{i,\pi(d)})}_{t_{\pi(d)}(\mathbf{x}_i)}.\text{''} \tag{2}$$

which contains $d$ permuted columns names and their corresponding values on the $i^{th}$ row. During training, we randomly select a permutation $\pi \sim S_d$ at each gradient step to ensure that the LLMs do not depend on the feature order. For inference, we fix the permutation for all test samples to maintain consistency.

## 2.3 Fine-tuning Large Language Models

Finally, we describe the fine-tuning process of a pretrained LLM with serialized tabular data $\mathcal{T} = \{\mathbf{T_c}(\pi, \mathbf{x}_i) \mid \pi \in S_d, i \in \{1, \cdots, n\}\}$. For each $\mathbf{s} \in \mathcal{T}$, the serialized tabular data is first transformed into a sequence of tokens, i.e. $\text{Tokenize}(\mathbf{s}) = (w_1, \cdots, w_{l(\mathbf{s})})$, of length $l(\mathbf{s})$ using the tokenizer of the given LLM. We then fine-tune the pretrained LLM in an auto-regressive manner using the causal language modeling loss:

$$\mathcal{L}_\theta = \mathbb{E}_{\mathbf{s} \in \mathcal{T}}[\ell_{clm}(\theta, \mathbf{s})] = \mathbb{E}_{\mathbf{s} \in \mathcal{T}}\left[-\sum_{k=1}^{l(\mathbf{s})} \log p_\theta(w_k | w_1, \cdots, w_{k-1})\right], \tag{3}$$

where $\theta$ is the parameters of the LLM. By fine-tuning a pretrained LLM with causal language modeling loss, it can learn to predict column values based on the columns that have already been seen. Given that pretrained LLMs have already acquired semantic understanding of the features, it can leverage its extensive contextual knowledge to predict the subsequent column values. For example, the features `age of vehicle` and `vehicle price` have clear coherence since a vehicle price decreases as it ages. Furthermore, by representing numerical feature values with decimal numbers, the pretrained LLMs can utilize their basic arithmetic abilities to encode quantity of numerical values (Hanna et al., 2024; Stolfo et al., 2023). This fine-tuning process not only reinforces the LLMs' comprehension of the specific formatting specified by our serialization method but also enhances their ability to learn the dependencies between different features.

## 2.4 Calculation of Anomaly Scores

Anomaly scores are calculated using the negative log-likelihood of the fine-tuned LLM, as this score reflects the (negative) probability of a sample being generated by the LLM. A higher negative log-likelihood indicates greater surprise from the LLM when predicting the inputs. Assuming that the majority of the training samples are normal, the fine-tuned LLM should predict normal inputs more accurately and assign higher probabilities to them. Since we found that directly computing the negative log-likelihood for textual features, which vary greatly in length, could introduce a length bias, we employ different methods to calculate anomaly scores.

**Case 1: Datasets comprised of only numerical and categorical columns:** We first discuss how to obtain probabilities of all categories in the categorical column $c_j$. Given the prefix sequence of the test sample $\mathbf{x}_i'$, "$c_1$ is $E(\mathbf{x}_{i,1}'), c_2$ is $E(\mathbf{x}_{i,2}'), \cdots, c_j$ is ", our objective is to obtain the conditional probabilities of all possible categories, given this prefix.

**Theorem 1.** *Given a prefix token sequence $s_0$, assume that there are only $q$ possible subsequent token sequences, which are $s^{(1)}, s^{(2)}, \cdots, s^{(q)}$. Let each sequence $s^{(i)}$ have probability $P(s^{(i)}|s_0)$ to be sampled. Let each sequence $s^{(i)}$ consisting of $l_i$ tokens, denoted as $w_1^{(i)}, w_2^{(i)}, \ldots, w_{l_i}^{(i)}$. Then the optimal language model $p_{\theta*}$ that minimizes the causal language modeling loss in Eqn.3 should align with the true distribution to satisfy the following equation:*

$$for\ all\ i \in \{1, 2, \cdots, q\}, p_{\theta*}(s^{(i)}|s_0) = \prod_{k=1}^{l_i} p_{\theta*}(w_k^{(i)}|s_0, w_1^{(i)}, ..., w_{k-1}^{(i)}) = P(s^{(i)}|s_0). \quad (4)$$

When we instantiate $s^{(1)}, s^{(2)}, \cdots, s^{(q)}$ with $q$ class of one category and $P(s^{(i)}|s_0)$ with their true conditional probabilities given the previously observed features, the above theorem shows that the product of all token probability resembles the probability of being in the corresponding category even though categorical names may have different lengths for the optimal LLMs. As LLMs have been fine-tuned with extensive tabular data using the same template as in Eqn.2, they should inherently recognize when to generate categorical data, the LLMs should meet the optimality conditions required for the above theorem to be applicable. Since we treat numerical columns as categorical, the theorem should also apply to numerical columns.

Note that we apply the same template, "$c_1$ is $E(\mathbf{x}'_{i,1}), c_2$ is $E(\mathbf{x}'_{i,2}), \cdots, c_j$ is ", across all test samples. Therefore, the token probabilities associated with the feature names, such as "$c_j$ is ", should be consistent across all test samples. As a result, we can compute anomaly scores by simply summing the log probabilities of every token in the sequence.

Next, given that the fine-tuned LLM has learned to generate tabular data *with arbitrary column orders*, we run the LLM inference $r$ times with different column orders in the test stage to reduce variance. Specifically, we compute anomaly scores using the following formula: given test samples $\mathbf{X}' = \{\mathbf{x}'_1, \cdots, \mathbf{x}'_m\}$, the anomaly score of the sample $\mathbf{x}'_i$ is given as follows:

$$\text{score}(\mathbf{x}'_i) = \frac{1}{r}\sum_{j=1}^{r}\ell_{clm}(\theta, \mathbf{T_c}(\pi_j, \mathbf{x}'_i)) = \frac{1}{r}\sum_{j=1}^{r}\sum_{k=1}^{l(\mathbf{T_c}(\pi_j, \mathbf{x}'_i))} -\log p_\theta(w'_k|w'_1, \cdots, w'_{k-1}), \quad (5)$$

where $\pi_1, \pi_2, \cdots, \pi_r \sim S_d$ are different permutations over $\{1, 2, \cdots, d\}$, $(w_1, \cdots, w_{l(\mathbf{T_c}(\pi_j, \mathbf{x}'_i))}) = \text{Tokenize}(\mathbf{T_c}(\pi_j, \mathbf{x}'_i))$ is the token sequence of the serialized tabular data $\mathbf{x}'_i$ with permutation $\pi_j$, and this sequence has length $l(\mathbf{T_c}(\pi_j, \mathbf{x}'_i))$. For consistency, we apply the same permutations $\pi_1, \pi_2, \cdots, \pi_r$ to all test samples. The impact of the number of permutations $r$ is analyzed in the experimental section.

**Case 2: Datasets containing textual columns:** For datasets containing textual features, we observe that Eqn.5 is heavily influenced by text lengths. Longer texts naturally yield higher anomaly scores due to the summation over more words. To address this, we normalize each textual column by its length separately and sum the scores up. Specifically, we compute anomaly scores using the formula, $\text{score}(\mathbf{x}'_i) = \frac{1}{r}\sum_{j=1}^{r}\sum_{k=1}^{d}\ell_{nor}(\theta, \pi_j, t_{\pi_j(k)}(\mathbf{x}'_i))$, with $\ell_{nor}$ defined below:

$$\ell_{nor}(\theta, \pi_j, t_{\pi_j(k)}(\mathbf{x}'_i)) = \frac{-1}{g(c_{\pi_j(k)})}\log p_\theta\left(E(\mathbf{x}'_{i,\pi_j(k)}) \mid "t_{\pi_j(1)}(\mathbf{x}'_i), \cdots, t_{\pi_j(k-1)}(\mathbf{x}'_i),\ c_{\pi_j(k)}\ is\ "\right).$$
$$(6)$$

When $c_{\pi_j(k)}$ is a textual column, we set $g(c_{\pi_j(k)})$ to the number of tokens in $E(\mathbf{x}'_{i,\pi_j(k)})$. When $c_{\pi_j(k)}$ is not a textual column, we set $g(c_{\pi_j(k)}) = 1$ so that it is not normalized by the number of tokens. Different from Eqn.5, we compute the negative log-likelihood only when predicting column values $E(\mathbf{x}'_{i,\pi_j(k)})$, and normalize the loss only for textual columns. We exclude the log-likelihood of the column name, "$c_{\pi_j(k)}$ is ", since the fine-tuned LLM recognizes it is as part of the serialization template and therefore should not be included when counting the text length. Normalization is done separately for each column to ensure that columns with longer texts do not disproportionately influence the anomaly scores.

## 3 EXPERIMENTS

**Datasets:** Since popular anomaly detection benchmarks, such as ADBench (Han et al., 2022) and the ODDS library (Rayana, 2016), mainly consist of numerical features, we manually col-

lect six datasets that contain mixed types of features. The six datasets are derived from ODDS library (Rayana, 2016), the fraud dataset benchmarks (Grover et al., 2022) and Kaggle. The dataset statistics are described in Table 1. To demonstrate the ability of AnoLLM to accommodate numerical columns, we also evaluate the approach on 30 datasets from the ODDS library, which are mainly composed of numerical features. The ODDS library is collected from various domains, such as chemistry, healthcare, and astronautics. Due to space constraints, we include dataset statistics with respect to the ODDS library in Table 4 of the Appendix.

| Datasets | # Samples | # text | # num | # category | # anomaly (%) |
|---|---|---|---|---|---|
| Fake job posts (Grover et al., 2022) | 17,880 | 5 | 3 | 8 | 866 (4.84%) |
| Fraud ecommerce (Grover et al., 2022) | 151,112 | 0 | 1 | 6 | 14,151 (9.36%) |
| Lymphography (Rayana, 2016) | 148 | 0 | 3 | 15 | 6 (4.05%) |
| Seismic (Rayana, 2016) | 2,584 | 0 | 14 | 4 | 170 (6.58%) |
| Vehicle insurance (Kaggle) | 15,420 | 0 | 8 | 24 | 923 (5.99%) |
| 20 newsgroup | 11,905 | 1 | 0 | 0 | 591 (4.96%) |

Table 1: Dataset statistics for the six datasets from the mixed-type benchmark. # text, # num, and # category stand for the numbers of textual, numerical, and categorical features in each dataset.

**Baselines:** We compare against 11 prominent methods in the field of tabular anomaly detection. For classical approaches, we compare against IForest (Liu et al., 2008), PCA (Shyu et al., 2003), KNN (Ramaswamy et al., 2000) and ECOD (Li et al., 2022). These are classical approaches that are still competitive against deep-learning-based methods (Han et al., 2022). For deep-learning based approaches, we compare against DeepSVDD (Ruff et al., 2018), RCA (Liu et al., 2021), and self-supervised learning based approaches including SLAD (Xu et al., 2023b), GOAD (Bergman & Hoshen, 2020), NeuTral (Qiu et al., 2021), ICL (Shenkar & Wolf, 2022), DTE (Livernoche et al., 2024) and REPEN (Pang et al., 2018). All methods are implemented with identical dataset partitioning.

**Implementation details:** We use `SmolLM-135M` and `SmolLM-360M` (Allal et al., 2024) as our backbone LLM since they represent state-of-the-art small models among open-weights LLMs. We choose small models (135M and 360M parameters) since the experimental results in Section 3.4 suggest that increasing model sizes does not provide much improvements. Fine-tuning is conducted for 2,000 steps with an AdamW optimizer (Loshchilov & Hutter, 2019) with learning rate $5 \times 10^{-5}$ across all datasets[1] as the training loss converges uniformly. Batch sizes are adjusted for each dataset to accommodate the varying lengths of serialized data. During inference, we select the number of permutations $r = 21$ since further increasing $r$ does not result in any observed improvement. Fine-tuning and inference are performed on seven Nvidia A100 40GB GPUs hosted on Amazon EC2 P4 Instances. Detailed hyperparameters are shown in Table 7 of the Appendix.

For baseline implementation, we use PyOD library (Zhao et al., 2019) for shallow methods and DeepOD library (Xu et al., 2023a) for deep-learning-based approaches. We standardize all numerical features to have a mean of zero and a standard deviation of one as AnoLLM does. For categorical features, we group all rare classes with less than 1% of samples together and use one-hot encoding[2]. For textual columns, we use averaged word2vec embeddings (Mikolov et al., 2013) across all words in each column. The word2vec embeddings are 300-dimensional and are trained from the Google news dataset [3]. For each method, we used the best-performing set of hyperparameters reported in their original paper. For others not specified, we use the default hyperparameters as suggested by DeepOD and PyOD toolkits.

**Evaluation protocols:** Following prior works (Shenkar & Wolf, 2022; Xu et al., 2023b), we conduct experiments in an uncontaminated, unsupervised setting. The training set consists of a random sample of 50% from the pool of normal examples, with the test set comprising the remaining normal examples, along with all anomalies. We randomly split each dataset using 5 different random seeds

---

[1]For the fake job post dataset, the model is fine-tuned for 10,000 steps due to a longer convergence time.

[2]For SLAD, we use ordinal encoding for categorical features as it does not support one-hot encoding.

[3]https://code.google.com/archive/p/word2vec/

| Methods \ Datasets | Fake job posts | Fraud ecommerce | Lympho-graphy | Seismic | Vehicle insurance | 20news groups | Average |
|---|---|---|---|---|---|---|---|
| Classical methods | | | | | | | |
| Iforest | 0.755 | 0.501 | 0.673 | 0.692 | 0.496 | 0.623 | 0.623 |
| PCA | 0.724 | 0.647 | 0.826 | 0.692 | 0.509 | 0.623 | 0.670 |
| KNN | 0.636 | **1** | 0.860 | 0.738 | 0.524 | 0.605 | 0.727 |
| ECOD | 0.512 | 0.755 | 0.830 | 0.692 | 0.509 | 0.62 | 0.653 |
| Deep learning based methods | | | | | | | |
| DeepSVDD | 0.561 | **1** | 0.899 | 0.713 | 0.505 | 0.597 | 0.713 |
| RCA | 0.629 | **1** | 0.919 | 0.727 | 0.531 | 0.546 | 0.725 |
| SLAD | 0.603 | 0.998 | 0.964 | 0.714 | 0.556 | 0.64 | 0.746 |
| GOAD | 0.566 | 0.998 | 0.817 | 0.717 | 0.512 | 0.63 | 0.707 |
| NeuTral | 0.548 | **1** | 0.847 | 0.681 | 0.507 | 0.658 | 0.707 |
| ICL | 0.699 | **1** | 0.827 | 0.719 | 0.501 | 0.671 | 0.736 |
| DTE | 0.548 | **1** | 0.909 | 0.714 | 0.512 | 0.6 | 0.714 |
| REPEN | 0.653 | **1** | 0.808 | 0.724 | 0.513 | 0.574 | 0.712 |
| AnoLLM | | | | | | | |
| SmolLM-135M | 0.800 | **1** | 0.968 | 0.712 | **0.569** | **0.766** | 0.803 |
| SmolLM-360M | **0.814** | **1** | **0.995** | **0.746** | 0.555 | 0.752 | **0.810** |

Table 2: AUC-ROC scores for all methods on the six datasets containing mixed types of features.

and reported the averaged results. We employ Area Under the Receiver Operating Characteristic Curve (AUC-ROC) as our primary evaluation metric. Other metrics, e.g. area under the precision-recall curves (AUC-PRC) and F1 scores, reflect similar trends in our experimental results and are shown in Section H of the Appendix. Moreover, a runtime comparison is shown in Section F of the Appendix.

## 3.1 MAIN RESULTS ON MIXED-TYPE TABULAR DATASETS

Table 2 shows the overall performance of different methods on the six mixed-type datasets. As shown in the table, AnoLLM consistently delivers the best results on all datasets. Specifically, AnoLLM with the SmolLM-360M backbone exhibits at least a 6.4% improvement over the baseline methods. This performance is especially notable on datasets with textual columns, such as the fake job posts and 20 news-groups datasets, where AnoLLM substantially outperforms other methods. This highlights the advantage of utilizing LLMs for anomaly detection, as they can process raw text inputs effectively and do not require the extensive feature engineering for textual inputs that is necessary for baseline methods. We also note that although AnoLLM's efficiency may be a concern due to its large model size, our observations indicate that, with the utilization of large GPU memory, the training time of AnoLLM-135M can be comparable to other deep learning-based methods, as discussed in Section F of the Appendix.

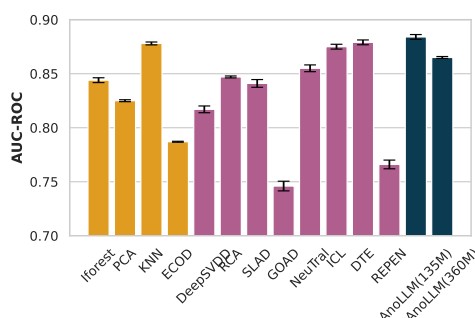

Figure 2: Averaged AUC-ROC scores with standard error bars for all methods over 30 datasets in ODDS. Colour scheme: yellow (shallow methods), purple (deep learning methods), dark blue (ours).

## 3.2 RESULTS ON THE ODDS BENCHMARK

Figure 2 presents the overall performance of these different methods on 30 tasks in the ODDS library. The results for each dataset and standard error are provided in Table 10 and Table 11 in Appendix. A critical difference diagram showing statistical significance is included as Figure 7 in the Appendix.

| LLM sizes | Mix-typed | ODDS |
|-----------|-----------|------|
| 135M | 0.803 | **0.884** |
| 360M | 0.811 | 0.865 |
| 1.7B | **0.812** | 0.861 |

(a) Comparison of different LLM sizes across the six mixed-type tabular benchmarks and the ODDS benchmark, with the numbers representing AUC-ROC scores.

| Methods | AUC-ROC |
|---------|---------|
| Equal-width | 0.865 |
| Quantile | 0.752 |
| Language | 0.863 |
| No binning | 0.800 |
| Standard | **0.884** |

(b) Comparison on different binning methods for numerical columns on ODDS library.

Table 3: Comparison of different LLM sizes and binning methods.

We note that in ODDS benchmarks, over $98.5\%$ of columns are numerical, $0\%$ of columns are textual, and only 10 out of 30 datasets contain human-understandable column names, which diminishes the strengths of LLMs. However, by using pseudo feature names and feature binning, AnoLLM with SmolLM-135M backbone performs comparably to the competitive baselines, KNN, ICL, and DTE. These results highlight AnoLLM's ability to effectively handle numerical data and its robustness to missing column names. These findings are consistent with previous studies on tabular anomaly detection which indicates that KNN, ICL, and DTE yield top performance (Livernoche et al., 2024).

## 3.3 EFFECTS OF FEATURE BINNING

In this experiment, we systematically study different options for feature binning strategies for numerical columns. Specifically, we choose to use the 30 datasets in ODDS library since they are primarily comprised of numerical columns. All methods use the same pretrained LLM, `smolLM-135M`. We compare the standard rescaling method as in Eqn.1 with four alternative methods listed below.

**Equal-width binning:** The range of each column is divided into 10 intervals, each with the same width. The resulting bins are labeled with single-digit decimal numbers, ranging from 0.1 to 1.0.

**Quantile binning:** All numerical columns are divided into 10 bins with each bin contain approximately $10\%$ of samples. The bins are labeled by percentiles, from "the 10th percentile" to "the 100th percentile".

**Language:** This method follows the same approach as equal-width binning, but the bin labels are replaced with 10 adjectives in natural language, representing magnitudes: "minimal", "slight", "moderate", "noticeable", "considerable", "significant", "substantial", "major", "extensive", and "maximum".

**No binning:** Numerical features are used in their raw and unaltered form.

Table 3b shows the averaged performance of each method over 30 datasets in the ODDS library. The results show that the standard rescaling method delivers the best performance, followed by equal-width binning and its language-based variant. In contrast, both no binning and quantile binning show comparatively lower performance, highlighting the importance of linear rescaling and rounding. This is evident as linear rescaling methods, such as equal-width binning and standard rescaling, outperform approaches that rely on raw numerical values, which may include long digit numbers that negatively impact overall performance. Moreover, replacing bin labels with adjectives has minimal effect on overall performance. Detailed comparison of each dataset can be seen in Table 16 in Appendix.

## 3.4 IMPACT OF LLM SIZES

In this experiment, we discuss the impact of LLM sizes to the AnoLLM framework. We compare LLMs in the `SmolLM` family as they share the same pre-training corpus and tokenizer, with the only difference being the number of model parameters [4]. Due to GPU memory constraints, we use a LoRA adapter (Hu et al., 2022) for the 1.7B model instead of full fine-tuning.

---

[4]We note that `SmolLM-1.7B` uses 1T tokens and 135M and 360M models only use 600M tokens for pre-training.

Table 3a shows the averaged performance for all LLM sizes on the mixed-type benchmark and the ODDS library. Detailed performance on each individual dataset is provided in Table 17 in the Appendix. The results suggest that using the 1.7B model does not provide much performance boost. This could be because larger models are trained on text data that are not relevant to our tabular tasks. As a result, the increased model capacity and contextual knowledge do not lead to any improvements. Therefore, we recommend that practitioners use smaller models, which offer a better balance of efficiency and effectiveness.

More ablation studies on the effects of random permutations and pretrained weights are provided in Appendix D.

## 4 Related Works

**Unsupervised Anomaly Detection for Tabular data:** Tabular anomaly detection is a long-standing problems and number of approaches, including classic methods and deep-learning based methods, are proposed over past several decades. One line of classical methods compute anomaly scores based on density, e.g. kernel density estimators (Latecki et al., 2007), local density (Breunig et al., 2000), and gaussian mixture models (Yang et al., 2009). Othe methods include isolation forest (Liu et al., 2008), empirical-cumulative-distribution-based outlier detector (ECOD) (Li et al., 2022), One-class support vector machine (Schölkopf et al., 1999), and k-nearest neighbors (KNN) (Ramaswamy et al., 2000). Notably, KNN remains a strong baseline even when numerous deep learning based approaches have been proposed (Livernoche et al., 2024).

Deep-learning-based approaches can be categorized into two groups, margin-based approaches and self-supervised learning (SSL) based approaches. Margin-based approaches, e.g. DeepSVDD (Ruff et al., 2018) and DROCC (Goyal et al., 2020), employ neural networks to map normal data into hyperspace with minimal volume. SSL-based approaches typically define pseudo-tasks where normal data is expected to perform better than abnormal data. As a result, these methods often incorporate SSL loss functions into their objectives. For example, NeuTral (Qiu et al., 2021) and REPEN (Pang et al., 2018) employ contrastive loss to bring the embeddings of similar data closer, while distancing the embeddings of dissimilar data. SLAD (Xu et al., 2023b) and ICL (Shenkar & Wolf, 2022) both construct pseudo tasks by splitting the input vectors. Please refer to Section G in Appendix for more details. We note that all the above methods are not able to handle raw texts and focus on numerical features only. Their applications to categorical data are not thoroughly discussed (Taha & Hadi, 2019).

**LLMs for Anomaly Detection:** With the rising popularity of LLMs, recent studies have explored their application in various anomaly detection (AD) tasks. In time-series AD, Liu et al. (2024a) demonstrate that LLMs can deliver accurate and interpretable results through well-designed prompts. Alnegheimish et al. (2024) investigate the zero-shot performance of LLMs in time-series AD. For image data, several studies (Gu et al., 2024; Cao et al., 2023; Zhu et al., 2024; Yang et al., 2024c) examine the performance of visual-linguistic models in visual and video AD across different settings. Additionally, Jin et al. (2024) and Liu et al. (2024b) show that large visual-linguistic models can effectively detect fake news using multi-modal inputs. For system log data, several studies (Han et al., 2023; Yamanaka et al., 2024; Lee et al., 2023b; Hadadi et al., 2024) have fine-tuned pretrained LLMs to detect anomalies in system logs.

For tabular data, a concurrent study (Li et al., 2024) examines the zero-shot performance of LLMs. Their approach differs from ours in two ways. First, they focus on zero-shot performance and report weaker results on the ODDS benchmark, whereas we fine-tune LLMs on tabular data, achieving better performance. Second, their method prompts LLMs to detect anomalies for each dimension separately and then aggregates the scores. Their performance lags behind ECOD, which also computes per-feature anomaly scores but does so by computing empirical cumulative distribution. In contrast, our methods are able to leverage contextual dependencies between features and achieve better results than ECOD. Additionally, Biester et al. (2024) and Park (2024) utilize LLMs as agents to generate domain-specific contexts or to format data, such as for financial data processing and data cleaning. Their methods require additional modules and, in some instances, need human intervention to process the outputs produced by the LLMs.

**LLMs for Tabular Data:**    Recent advances in LLMs have spurred extensive exploration of their use in various tabular data tasks, including prediction (Dinh et al., 2022; Hegselmann et al., 2023; Manikandan et al., 2023), synthesis (Borisov et al., 2023; Zhang et al., 2023), feature engineering (Han et al., 2024), and table understanding (Sui et al., 2024). Additionally, recent studies have investigated pre-training foundation models specifically for tabular data (Zhu et al., 2023; Yan et al., 2024; Ye et al., 2024; Yang et al., 2024b;a). For a comprehensive overview, we refer readers to a survey paper (Fang et al., 2024).

## 5    CONCLUSION AND FUTURE WORK

We introduce AnoLLM, a novel framework for adapting pretrained LLMs to unsupervised anomaly detection for tabular data. AnoLLM is a robust tabular anomaly detection method that can inget raw textual features without pre-processing. Empirical evaluations demonstrate that AnoLLM achieves state-of-the-art performance across six benchmark datasets containing mixed feature types, and matches the performance of the leading methods, KNN, ICL, and DTE, on 30 datasets predominantly containing numerical attributes.

One future direction for this work could be reducing the computational overhead associated with AnoLLM, given that LLMs are much less efficient compared to traditional anomaly detection methods. Furthermore, considering AnoLLM's superior capability in modeling categorical and textual features, an interesting direction for future work could involve utilizing LLMs to extract salient representations from such features, which could then be leveraged by classical anomaly detection methods to improve efficiency. Furthermore, since LLMs are known to exhibit limitations in numerical reasoning, it is of interest to enhance AnoLLM's performance on numerical data. One potential improvement could involve the development of specialized tokenization strategies and encoding tailored to numerical attributes.

Finally, as building foundation models for tabular data is an ongoing trend (Van Breugel & Van Der Schaar, 2024; Gardner et al., 2024), AnoLLM represents an initial step in this direction by demonstrating that LLMs can be effectively adapted for tabular anomaly detection. While numerous other tabular tasks remain to be explored, an important avenue for future work is the creation of a general-purpose tabular foundation model capable of addressing a wide range of tabular data challenges.

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

## A  OVERVIEW OF THE APPENDIX

In Section B, we present detailed information about the datasets used in our experiments. Section C analyzes AnoLLM's sensitivity to the number of permutations $r$. Section D includes more ablation studies on the effects of random pertmuations and pretrained weights. The proof of Theorem 1 is provided in Section E. Section F discusses the computational efficiency of our approach. A comprehensive review of related work on tabular anomaly detection is included in Section G. Finally, Section H details the hyperparameters of AnoLLM and provides full experimental results, including additional performance metrics.

## B  DETAILED DATASET INFORMATION

In this section, we provide detailed dataset information that we used in our experiments. The datasets statistics of the seven main datasets are in Table 1.

**Fake job posts (Grover et al., 2022):**  This Kaggle dataset comprises 18,000 job descriptions, approximately 800 of which are fraudulent. The dataset has 5 textual features that are related to job postings, which are title, company profile, job description, requirements, and benefits. The objective is to develop a classification model capable of identifying fraudulent job postings. We download the dataset using the provided API [5].

**Fraud ecommerce (Grover et al., 2022):**  This dataset comprises 150,000 e-commerce transactions, featuring data such as sign-up time, purchase time, purchase value, device ID, browser, and IP address. We use the provided table to convert IP addresses into the countries and discard the device ID feature. We also follow (Grover et al., 2022) to use the time difference between sign-up and purchase. We download the dataset using the provided API.

**Lymphography (Rayana, 2016):**  We collect this dataset from the ODDS library (Rayana, 2016), but it was originally obtained from the University Medical Centre, Institute of Oncology, Ljubljana. The dataset features 18 attributes with the primary objective of predicting whether a lymphography result is normal. The objective is to predict whether high-energy seismic bump will occur in the next shift.

**Seismic (Rayana, 2016):**  The dataset addresses the issue of forecasting seismic bumps with high energy levels (greater than $10^4$ J) in a coal mine. The data were collected from two longwalls in a Polish coal mine.

**Vehicle insurance**  : The dataset comprises 32 metadata of claimants and their insurance details, including 6 ordinal and 25 categorical. It contains 15,420 records, with only 6% (923 records) identified as fraudulent. The objective is to predict whether a claim is fraudulent. The dataset is publically available and is provided by Angoss Knowledge Seeker. We collect the dataset from Kaggle [6].

**20 newsgroups:**  The dataset comprises news articles from 20 distinct newsgroups [7]. Following the configuration outlined in ADBench (Han et al., 2022), we use six top-level categories: computer, recreation, science, miscellaneous, politics, and religion. We conduct six experiments, each time designating one of the six classes as the normal class and treating the remaining classes as anomalies. The anomalous classes are downsampled to represent 5% of the total instances. The average performance of the six experiments is reported.

**Outlier Detection DataSets (ODDS) (Rayana, 2016):**  The ODDS provides datasets from different domains, e.g. healthcare, image, finance, botany, etc, each with varying numbers of features and

---

[5]https://github.com/amazon-science/fraud-dataset-benchmark

[6]https://www.kaggle.com/datasets/khusheekapoor/vehicle-insurance-fraud-detection

[7]https://kdd.ics.uci.edu/databases/20newsgroups/20newsgroups.data.html

samples. The statistics are provided in Table 4. We note that ODDS is mainly comprised of numerical columns as over $98.5\%$ of columns are numerical. Also, for datasets lacking explicit column names, we use pseudo column names, e.g. "$A$", "$B$", ...,"$Z$", "$AA$", ....

| Dataset | # points | # text | # num | # category | Has column names | # outliers (%) |
|---|---|---|---|---|---|---|
| Annthyroid | 7,200 | 0 | 6 | 0 | No | 534 (7.42%) |
| Arrhythmia | 452 | 0 | 274 | 0 | No | 66 (15%) |
| BreastW | 683 | 0 | 9 | 0 | Yes | 239 (35%) |
| Cardio | 1,831 | 0 | 21 | 0 | Yes | 176 (9.6%) |
| Ecoli | 336 | 0 | 7 | 0 | Yes | 9 (2.6%) |
| ForestCover | 286,048 | 0 | 10 | 0 | Yes | 2,747 (0.9%) |
| Glass | 214 | 0 | 9 | 0 | No | 9 (4.2%) |
| Heart | 224 | 0 | 44 | 0 | No | 10 (4.4%) |
| Http (KDDCUP99) | 567,479 | 0 | 3 | 0 | No | 2,211 (0.4%) |
| Ionosphere | 351 | 0 | 33 | 0 | No | 126 (36%) |
| Letter Recognition | 1,600 | 0 | 32 | 0 | No | 100 (6.25%) |
| Lymphography | 148 | 0 | 3 | 15 | Yes | 6 (4.1%) |
| Mammography | 11,183 | 0 | 6 | 0 | No | 260 (2.32%) |
| Mulcross | 262,144 | 0 | 4 | 0 | No | 26,214 (10%) |
| Musk | 3,062 | 0 | 166 | 0 | No | 97 (3.2%) |
| Optdigits | 5,216 | 0 | 64 | 0 | No | 150 (3%) |
| Pendigits | 6,870 | 0 | 16 | 0 | No | 156 (2.27%) |
| Pima | 768 | 0 | 8 | 0 | No | 268 (35%) |
| Satellite | 6,435 | 0 | 36 | 0 | No | 2,036 (32%) |
| Satimage-2 | 5,803 | 0 | 36 | 0 | No | 71 (1.2%) |
| Seismic | 2,584 | 0 | 14 | 4 | Yes | 170 (6.5%) |
| Shuttle | 49,097 | 0 | 9 | 0 | No | 3,511 (7%) |
| Smtp (KDDCUP99) | 95,156 | 0 | 3 | 0 | No | 30 (0.03%) |
| Speech | 3,686 | 0 | 400 | 0 | No | 61 (1.65%) |
| Thyroid | 3,772 | 0 | 6 | 0 | No | 93 (2.5%) |
| Vertebral | 240 | 0 | 6 | 0 | Yes | 30 (12.5%) |
| Vowels | 1,456 | 0 | 12 | 0 | No | 50 (3.4%) |
| WBC | 278 | 0 | 30 | 0 | No | 21 (5.6%) |
| Wine | 129 | 0 | 13 | 0 | Yes | 10 (7.7%) |
| Yeast | 1,364 | 0 | 8 | 0 | Yes | 64 (4.7%) |

Table 4: Summary of datasets in Outlier Detection DataSets (ODDS) (Rayana, 2016).

## C    SENSITIVITY ANALYSIS OF THE NUMBER OF PERMUTATIONS

In this section, we explore the sensitivity of AnoLLM's performance to the number of permutations $r$ during inference. Figure 3 illustrates how performance changes with varying values of $r$. The blue shaded area indicates the standard deviation, calculated from bootstrapping with 100 trials for each $r$. The AnoLLM uses SmolLM-135M as the LLM backbone.

As shown in the figure, the average performance plateaus after $r = 10$ and it has roughly $0.8\%$ gain compared to $r = 1$. Additionally, the standard deviation decreases as $r$ increases. The results indicate that utilizing multiple permutations during inference enhances performance and reduces variance as more anomaly scores are ensembled from different permutation functions. Practitioners can select the number of permutations $r$ by considering the tradeoff between effectiveness and efficiency, as illustrated in this figure. We choose $r = 21$ in our experiments as it provides the best performance.

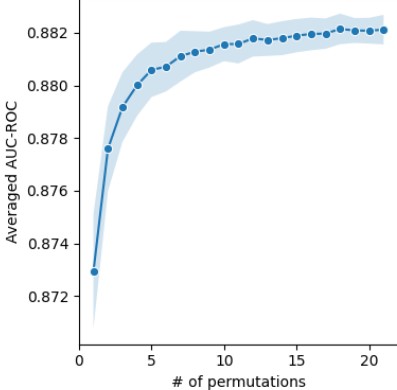

Figure 3: The average change of AnoLLM in AUC-ROC with respect to the number of permutations $r$ on 30 datasets from the ODDS library. The blue shaded area represents the standard deviation. The AnoLLM uses SmolLM-135M as the LLM backbone.

# D   MORE ON ABLATION STUDIES

In this section, we study the effect of random permutation and the effect of pretrained weights of LLMs.

## D.1   EFFECT OF RANDOM PERMUTATION

To study the effect of random permutations to the final performance, we train another SmolLM-135M on the five datasets containing mixed-typed features [8] using the same hyperparameters. We maintain the original column order from the dataset during both the training and inference stages. As demonstrated in Table 5, removing random permutations leads to a significant decline in AnoLLM's performance. This finding underscores the importance of random permutations in reducing the model's sensitivity to column ordering.

| Methods \ Datasets | Fake job posts | Fraud ecommerce | Lympho-graphy | Seismic | Vehicle insurance | Average |
|---|---|---|---|---|---|---|
| SmolLM-135M (w permutation) | **0.800** | **1** | **0.968** | **0.712** | 0.569 | **0.809** |
| SmolLM-135M (w.o. permutation) | 0.565 | 0.995 | 0.656 | 0.438 | **0.577** | 0.646 |

Table 5: AUC-ROC scores for AnoLLMs with and without random permutation on five datasets containing mixed typed features.

## D.2   EFFECT OF PRETRAINED WEIGHTS

To study the performance of AnoLLMs without LLM pretrained weights, we evaluate the performance of randomly initialized transformers on the ODDS benchmark. To ensure a fair comparison, we use the same model architecture, SmolLM-135M, with identical hyperparameters. The results are summarized in Table 6.

As shown, AnoLLM with pretrained weights slightly outperforms its randomly initialized counterpart in terms of overall average performance. It achieves better performance on 16 out of 30 datasets and matches performance on 4 datasets. Additionally, a visual inspection of the training curves reveals that AnoLLM with pretrained weights converges approximately twice as fast on most datasets. This faster convergence can be attributed to the pretrained LLM providing a better initialization for fine-tuning. In contrast, AnoLLM without pretrained weights not only converges more slowly but is also more susceptible to overfitting, as evidenced by its significantly lower training loss. While overfitting may not be a major concern in uncontaminated, unsupervised settings, it could present challenges in contaminated scenarios, where the model risks memorizing anomalous samples.

Given that the performance gap between variants is relatively modest, the randomly initialized AnoLLM can serve as a viable alternative for datasets primarily composed of numerical attributes in uncontaminated settings, particularly when smaller pretrained models are unavailable. An interesting direction for future work would be exploring the trade-off between efficiency and accuracy.

---

[8]We exclude the 20 newsgroups dataset because it includes only a single text column, making no difference for random column permutation.

| | Pretrained LLM weights | Randomly initialized weights |
|---|---|---|
| Annthyroid | 0.927 | **0.93** |
| Arrhythmia | 0.825 | **0.827** |
| BreastW | 0.992 | **0.993** |
| Cardio | **0.94** | 0.935 |
| Ecoli | 0.777 | **0.778** |
| ForestCover | **0.881** | 0.853 |
| Glass | **0.819** | 0.816 |
| Heart | **0.82** | 0.825 |
| Http (KDDCUP99) | **1** | **1** |
| Ionosphere | **0.909** | 0.89 |
| Letter Recognition | **0.967** | 0.907 |
| Lymphography | 0.968 | **0.997** |
| Mammography | **0.915** | 0.878 |
| Mulcross | **1** | **1** |
| Musk | **1** | **1** |
| Optdigits | **0.983** | 0.897 |
| Pendigits | 0.971 | **0.988** |
| Pima | **0.663** | 0.649 |
| Satellite | **0.902** | 0.86 |
| Satimage-2 | **1** | 0.998 |
| Seismic | 0.712 | **0.737** |
| Shuttle | **1** | **1** |
| Smtp (KDDCUP99) | **0.927** | 0.926 |
| Speech | **0.47** | 0.459 |
| Thyroid | 0.975 | **0.984** |
| Vertebral | **0.565** | 0.415 |
| Vowels | **0.982** | 0.895 |
| WBC | **0.964** | 0.953 |
| Wine | **0.909** | 0.884 |
| Yeast | 0.744 | **0.749** |
| Average | **0.884** | 0.867 |

Table 6: Comparison of AnoLLMs with randomly initialized weights and pretrained LLM weights (SmolLM-135M) on the ODDS benchmark.

# E    PROOF OF THEOREM 1

*Proof.* For all $w_1, w_2..., w_k$, we have

$$
\begin{aligned}
p_{\theta^*}(w_k|s_0, w_1, ..., w_{k-1}) &= \frac{p_{\theta^*}(w_1, ..., w_k|s_0)}{p_{\theta^*}(w_1, \cdots, w_{k-1}|s_0)} \\
&= \frac{\mathbb{E}_{s^{(i)} \sim P}\left[\mathbb{1}(\text{the first } k \text{ tokens of } s^{(i)} \text{ are } w_1, ..., w_k)|s_0\right]}{\mathbb{E}_{s^{(i)} \sim P}\left[\mathbb{1}(\text{the first } k-1 \text{ tokens of } s^{(i)} \text{ are } w_1, ..., w_{k-1})|s_0\right]},
\end{aligned}
$$

where the second equation is since the optimal language model $p_{\theta^*}$ aligns with the true distribution of $s^{(i)}$ being sampled. Then, for all $i \in \{1, 2, \cdots, q\}$, we have

$$
\begin{aligned}
&\prod_{k=1}^{l_i} p_{\theta^*}(w_k^{(i)}|s_0, w_1^{(i)}, ..., w_{k-1}^{(i)}) \\
&= \prod_{k=1}^{l_i} \frac{\mathbb{E}_{s^{(i)} \sim P}\left[\mathbb{1}(\text{the first } k \text{ words of } s^{(i)} \text{ are } w_1^{(i)}, ..., w_k^{(i)})|s_0\right]}{\mathbb{E}_{s^{(i)} \sim P}\left[\mathbb{1}(\text{the first } k-1 \text{ words of } s^{(i)} \text{ are } w_1^{(i)}, ..., w_{k-1}^{(i)})|s_0\right]} \\
&= \mathbb{E}_{s^{(i)} \sim P}\left[\mathbb{1}(\text{the first } l_i \text{ words of } s^{(i)} \text{ are } w_1^{(i)}, ..., w_{l_i}^{(i)})|s_0\right] \\
&= P(s^{(i)}|s_0).
\end{aligned}
$$

$\square$

## F    COMPUTE EFFICIENCY ANALYSIS

The total compute required to train AnoLLM-135M across all datasets with five seeds, including six datasets from the mixed-type benchmark and 30 datasets from the ODDS benchmark, is approximately 90 GPU hours on a single RTX-A6000 GPU with 48 GB of memory.

Figure 4 and Figure 5 illustrate the training and inference times for all methods, averaged over five datasets (fake job posts, fraud ecommerce, lymphography, seismic, and vehicle insurance). As anticipated, deep learning-based methods exhibit significantly higher training times and slightly higher inference time compared to classical methods. One exception is that KNN is much slower during inference since fraud ecommerce contains over $150,000$ training samples, making naive search computationally slow. Notably, AnoLLM-135M achieves comparable training times due to the large GPU memory of the RTX-A6000, which enables efficient batch processing. However, it is considerably slower during inference due to its large model size. We note that we use $r = 21$ different permutations throughout the paper, but the number can be reduced by selecting a smaller $r$. The trade-off between number of permutations $r$ and performance is shown in Fig.3.

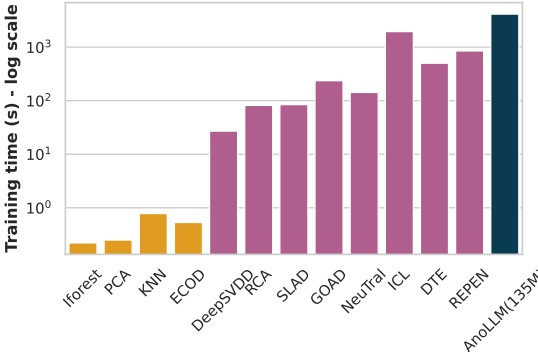

Figure 4: Mean training time over 5 datasets, fake job posts, fraud ecommerce, lymphography, seismic, and vehicle insurance. Colour scheme: yellow (shallow methods), purple (deep learning methods), dark blue (ours).

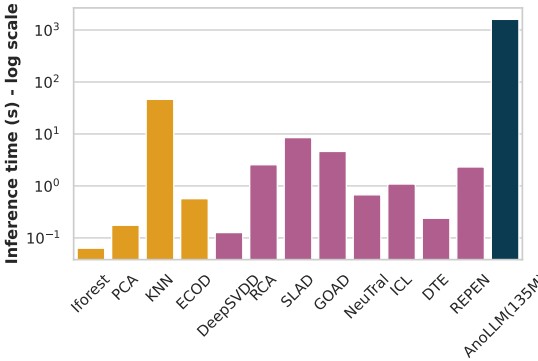

Figure 5: Mean inference time over 5 datasets, fake job posts, fraud ecommerce, lymphography, seismic, and vehicle insurance. Colour scheme: yellow (shallow methods), purple (deep learning methods), dark blue (ours).

# G   MORE RELATED WORK IN UNSUPERVISED ANOMALY DETECTION FOR TABULAR DATA

Tabular anomaly detection is a long-standing problems and a number of approaches, including classic methods and deep-learning based methods, are proposed over past several decades. One line of classical methods computes anomaly scores based on density, e.g. kernel density estimators (Latecki et al., 2007), local density (Breunig et al., 2000), and gaussian mixture models (Yang et al., 2009). Isolation forest (Liu et al., 2008) uses recursive partitioning with binary trees to identify outliers. Empirical-cumulative-distribution-based outlier detector (ECOD) (Li et al., 2022) estimates tail probabilities per dimension by computing empirical distributions and aggregates these probabilities to detect anomalies. One-class support vector machine (Schölkopf et al., 1999) finds the best boundary that separates data points into normal and anomalous classes. The k-nearest neighbors (KNN) (Ramaswamy et al., 2000) uses the distance to the $k^{th}$ neighbors as its anomaly scores, which remains a strong baseline (Livernoche et al., 2024).

Deep-learning-based approaches can be categorized into two groups, margin-based approaches and self-supervised learning (SSL) based approaches. Margin-based approaches, e.g. DeepSVDD (Ruff et al., 2018) and DROCC (Goyal et al., 2020), employ neural networks to map normal data into hyperspace with minimal volume. Deep isolation forest (Xu et al., 2023a) introduces a new representation scheme learned from neural networks to perform data partition. SSL-based approaches typically define pseudo-tasks where normal data is expected to perform better than abnormal data. As a result, these methods often incorporate SSL loss functions into their objectives. For example, NeuTral (Qiu et al., 2021) employs contrastive loss to bring the embeddings of augmented data closer to those of the original samples, while simultaneously pushing away the embeddings of augmented data derived from the same inputs. GOAD (Bergman & Hoshen, 2020) predefines a set of fixed transformations and trains a classifier to differentiate between them. REPEN (Pang et al., 2018) uses triplet loss to encourage normal samples to be closer than outlier samples that are generated by classical approaches. RCA (Liu et al., 2021) uses two auto-encoders to filter training samples with high loss and uses reconstruction loss as its anomaly scores. SLAD (Xu et al., 2023b) and ICL (Shenkar & Wolf, 2022) both split the input vectors. SLAD learns a model to predict the number of dimensions given the split vectors and ICL learns an embedding model to map vectors from the same input to nearby regions. MCM (Yin et al., 2024) utilizes masked prediction loss with learnable masks to detect anomalies. RDP (Wang et al., 2021) trains neural networks to predict the random distance of samples. DTE (Livernoche et al., 2024) estimates diffusion time and uses it as anomaly scores. Non-Parametric Transformers (Thimonier et al., 2024) (NPEs) detect anomalies in tabular data by reconstructing masked features of normal samples. We note that all the above methods are not able to handle raw textual columns and focus on numerical features only. Their applications to categorical data are not thoroughly discussed (Taha & Hadi, 2019).

## H  FULL RESULTS

In this section, we present the hyperparameters we used for AnoLLMs and the detailed performance of each method across all datasets. The hyperparameters of AnoLLMs are shown in Table 7. For mixed-typed benchmark, a critical difference diagram computed based on AUC-ROC scores is shown in Fig 6. Table 8 and Table 9 shows the F1 and area under the precision-recall curve (AUC-PR) scores for the six datasets containing mixed types of features.

For the ODDS benchmark, the detailed AUC-ROC scores, their standard errors, and critical difference diagrams for all methods are shown from Table 10 to Table 11 and Fig 7. The F1 and AUC-PR scores and their standard errors for all methods are shown in Table 12 and Table 14 Table 16 and Table 17 present the detailed performance of experiments in comparing different binning methods and LLM sizes. These results complement those presented in the experiments.

| Datasets \ Hyperparameters | SmolLM-135M | | SmolLM-360M | | SmolLM-1.7B | |
|---|---|---|---|---|---|---|
| | batch size | Use LoRA | batch size | Use LoRA | batch size | Use LoRA |
| Mix-type Benchmark | | | | | | |
| fake job post | 3 | Yes | 12 | Yes | 4 | Yes |
| fraud ecommerce | 32 | No | 32 | No | 16 | Yes |
| Lymphography | 32 | No | 32 | No | 16 | Yes |
| Seismic | 16 | No | 16 | No | 8 | Yes |
| Vehicle insurance | 32 | No | 32 | No | 16 | Yes |
| 20 newsgroups | 16 | No | 16 | No | 4 | Yes |
| ODDS Benchmark | | | | | | |
| Annthyroid | 16 | No | 16 | No | 4 | Yes |
| Arrhythmia | 2 | Yes | 2 | Yes | 1 | Yes |
| BreastW | 32 | No | 32 | No | 8 | Yes |
| Cardio | 32 | No | 32 | No | 8 | Yes |
| Ecoli | 32 | No | 32 | No | 8 | Yes |
| ForestCover | 48 | No | 48 | No | 12 | Yes |
| Glass | 32 | No | 32 | No | 8 | Yes |
| Heart | 32 | No | 32 | No | 8 | Yes |
| Http (KDDCUP99) | 128 | No | 128 | No | 32 | Yes |
| Ionosphere | 32 | No | 32 | No | 8 | Yes |
| Letter Recognition | 32 | No | 32 | No | 8 | Yes |
| Lymphography | 32 | No | 32 | No | 8 | Yes |
| Mammography | 32 | No | 32 | No | 8 | Yes |
| Mulcross | 96 | No | 96 | No | 24 | Yes |
| Musk | 8 | Yes | 8 | Yes | 2 | Yes |
| Optdigits | 16 | No | 16 | No | 4 | Yes |
| Pendigits | 32 | No | 32 | No | 8 | Yes |
| Pima | 32 | No | 32 | No | 8 | Yes |
| Satellite | 32 | No | 32 | No | 8 | Yes |
| Satimage-2 | 32 | No | 32 | No | 8 | Yes |
| Seismic | 16 | No | 16 | No | 8 | Yes |
| Shuttle | 32 | No | 32 | No | 8 | Yes |
| Smtp (KDDCUP99) | 128 | No | 128 | No | 32 | Yes |
| Speech | 2 | Yes | 2 | Yes | 1 | Yes |
| Thyroid | 32 | No | 32 | No | 8 | Yes |
| Vertebral | 32 | No | 32 | No | 8 | Yes |
| Vowels | 32 | No | 32 | No | 8 | Yes |
| WBC | 32 | No | 32 | No | 8 | Yes |
| Wine | 32 | No | 32 | No | 8 | Yes |
| Yeast | 32 | No | 32 | No | 8 | Yes |

Table 7: Hyperparameters used in the experiments for AnoLLMs with different backbones. For LoRA (Hu et al., 2022), we use the default parameters, $r = 16$ and $\alpha = 32$ and apply to all modules including all attention and MLP layers.

| Methods \ Datasets | Fake job posts | Fraud ecommerce | Lympho-graphy | Seismic | Vehicle insurance | 20news groups | Average |
|---|---|---|---|---|---|---|---|
| Classical methods | | | | | | | |
| Iforest | **0.274** | 0.173 | 0.233 | 0.251 | 0.11 | 0.137 | 0.196 |
| PCA | 0.256 | 0.209 | 0.567 | 0.266 | 0.124 | 0.133 | 0.259 |
| KNN | 0.163 | **1** | 0.667 | 0.291 | 0.135 | 0.156 | 0.402 |
| ECOD | 0.165 | 0.408 | 0.400 | 0.282 | 0.112 | 0.132 | 0.250 |
| Deep learning based methods | | | | | | | |
| DeepSVDD | 0.136 | **1** | 0.567 | 0.258 | 0.115 | 0.152 | 0.371 |
| RCA | 0.137 | **1** | 0.667 | 0.32 | 0.135 | 0.129 | 0.398 |
| SLAD | 0.175 | 0.988 | 0.667 | 0.285 | 0.155 | 0.159 | 0.405 |
| GOAD | 0.129 | 0.92 | 0.667 | 0.295 | 0.119 | 0.136 | 0.378 |
| NeuTral | 0.115 | **1** | 0.633 | 0.195 | 0.12 | 0.195 | 0.376 |
| ICL | 0.245 | **1** | 0.667 | 0.298 | 0.108 | 0.19 | 0.418 |
| DTE | 0.107 | **1** | 0.667 | 0.239 | 0.121 | 0.185 | 0.387 |
| REPEN | 0.164 | **1** | 0.667 | 0.306 | 0.126 | 0.124 | 0.398 |
| AnoLLM | | | | | | | |
| SmolLM-135M | 0.325 | **1** | 0.767 | 0.279 | 0.162 | **0.241** | 0.462 |
| SmolLM-360M | **0.343** | 0.992 | **0.8** | **0.336** | **0.174** | 0.22 | **0.478** |

Table 8: Detailed F1 scores for all methods on the six datasets containing mixed types of features. This table complements the results shown in Section 3.1. The scores are averaged over 5 random dataset splits.

| Methods \ Datasets | Fake job posts | Fraud ecommerce | Lympho-graphy | Seismic | Vehicle insurance | 20news groups | Average |
|---|---|---|---|---|---|---|---|
| Classical methods | | | | | | | |
| Iforest | 0.227 | 0.172 | 0.232 | 0.235 | 0.112 | 0.146 | 0.187 |
| PCA | 0.194 | 0.238 | 0.624 | 0.216 | 0.117 | 0.143 | 0.255 |
| KNN | 0.138 | **1** | 0.720 | 0.256 | 0.123 | 0.148 | 0.398 |
| ECOD | 0.13 | 0.39 | 0.365 | 0.244 | 0.116 | 0.141 | 0.231 |
| Deep learning based methods | | | | | | | |
| DeepSVDD | 0.12 | **1** | 0.680 | 0.226 | 0.114 | 0.135 | 0.379 |
| RCA | 0.134 | **1** | 0.783 | 0.25 | 0.124 | 0.136 | 0.405 |
| SLAD | 0.15 | 0.992 | 0.795 | 0.241 | 0.14 | 0.159 | 0.413 |
| GOAD | 0.117 | 0.979 | 0.697 | 0.239 | 0.116 | 0.144 | 0.382 |
| NeuTral | 0.108 | **1** | 0.681 | 0.193 | 0.117 | 0.176 | 0.379 |
| ICL | 0.193 | **1** | 0.718 | 0.251 | 0.115 | 0.175 | 0.452 |
| DTE | 0.106 | **1** | 0.747 | 0.224 | 0.116 | 0.157 | 0.392 |
| REPEN | 0.146 | **1** | 0.697 | 0.249 | 0.118 | 0.126 | 0.389 |
| AnoLLM | | | | | | | |
| SmolLM-135M | 0.286 | 0.999 | 0.856 | 0.236 | 0.141 | **0.223** | 0.457 |
| SmolLM-360M | **0.304** | 0.972 | **0.938** | **0.281** | **0.143** | 0.214 | **0.475** |

Table 9: Detailed AUC-PR scores for all methods on the six datasets containing mixed types of features. This table complements the results shown in Section 3.1. The scores are averaged over 5 random dataset splits.

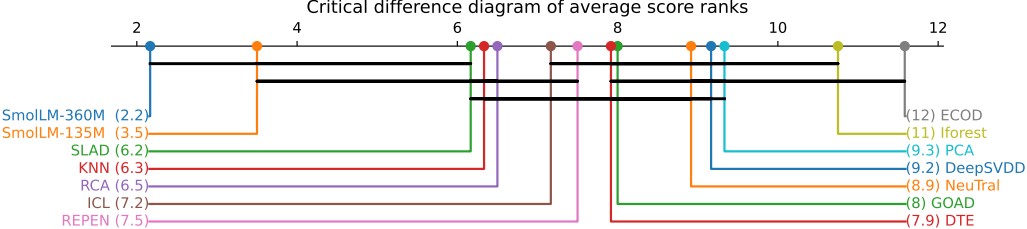

Figure 6: Critical difference diagram of average ranks for various methods. The ranks are computed over AUC-ROC scores. The x-axis shows the average ranks across datasets, with smaller values indicating better performance. Horizontal bars connect groups of methods that are not significantly different in performance according to a statistical test. We note that AnoLLMs are not significantly better than SLAD in terms of ranking since the ranking score is only averaged over six datasets. However, it outperforms it by a large margin as can be shown in Table 2.

| Datasets \Methods | Classical Methods | | | | Deep-learning based methods | | | | | | | | AnoLLM | |
|---|---|---|---|---|---|---|---|---|---|---|---|---|---|---|
| | Iforest | PCA | KNN | ECOD | DeepSVDD | RCA | SLAD | GOAD | NeuTral | ICL | DTE | REPEN | 135M | 360M |
| Annthyroid | 0.922 | 0.839 | 0.811 | 0.79 | 0.742 | 0.718 | 0.761 | 0.572 | 0.813 | 0.842 | **0.977** | 0.736 | 0.927 | **0.931** |
| Arrhythmia | **0.827** | 0.796 | 0.786 | 0.811 | 0.765 | 0.786 | 0.784 | 0.681 | 0.76 | 0.785 | 0.771 | 0.684 | **0.825** | 0.822 |
| BreastW | **0.994** | 0.988 | 0.992 | 0.992 | 0.974 | 0.987 | 0.986 | **0.994** | 0.983 | 0.992 | 0.982 | 0.955 | 0.992 | 0.993 |
| Cardio | **0.948** | **0.966** | 0.921 | 0.935 | 0.842 | **0.948** | 0.84 | 0.524 | 0.859 | 0.894 | 0.92 | 0.829 | 0.94 | 0.873 |
| Ecoli | 0.856 | 0.856 | 0.877 | 0.776 | **0.887** | 0.883 | 0.882 | 0.881 | 0.86 | **0.887** | 0.821 | 0.87 | 0.777 | 0.804 |
| ForestCover | 0.87 | 0.945 | **0.985** | 0.921 | 0.533 | 0.944 | 0.857 | 0.278 | 0.898 | 0.977 | **0.978** | 0.902 | 0.881 | 0.835 |
| Glass | 0.802 | 0.713 | 0.849 | 0.693 | 0.824 | 0.719 | 0.79 | 0.574 | **0.933** | **0.887** | 0.799 | 0.755 | 0.819 | 0.797 |
| Heart | 0.819 | **0.842** | 0.814 | 0.659 | 0.773 | 0.785 | 0.822 | **0.838** | 0.811 | 0.787 | 0.831 | 0.449 | 0.82 | 0.799 |
| Http (KDDCUP99) | 0.992 | 0.999 | 1 | 0.979 | 0.99 | 0.995 | 0.999 | 0.996 | 0.973 | 1 | 0.995 | 0.994 | 1 | 1 |
| Ionosphere | 0.891 | 0.894 | 0.96 | 0.734 | 0.963 | 0.916 | 0.96 | 0.95 | 0.956 | **0.97** | **0.964** | 0.545 | 0.909 | 0.924 |
| Letter Recognition | 0.631 | 0.529 | 0.865 | 0.567 | 0.776 | 0.716 | 0.908 | 0.811 | 0.929 | **0.959** | 0.872 | 0.597 | **0.967** | 0.867 |
| Lymphography | 0.673 | 0.826 | 0.86 | 0.83 | 0.899 | 0.919 | 0.964 | 0.817 | 0.847 | 0.827 | 0.909 | 0.808 | **0.968** | **0.993** |
| Mammography | 0.881 | 0.9 | 0.872 | **0.906** | 0.857 | 0.873 | 0.74 | 0.756 | 0.69 | 0.782 | 0.864 | 0.863 | **0.915** | 0.876 |
| Mulcross | 0.999 | 1 | 1 | 0.96 | 1 | 1 | 0.969 | 1 | 0.968 | 1 | 1 | 0.973 | 1 | 1 |
| Musk | 0.971 | 1 | 1 | 0.956 | 1 | 1 | 1 | 1 | 1 | 1 | 1 | 0.722 | 1 | 1 |
| Optdigits | 0.824 | 0.574 | 0.944 | 0.606 | 0.753 | 0.8 | 0.735 | 0.847 | **0.979** | 0.955 | 0.888 | 0.607 | **0.983** | 0.939 |
| Pendigits | 0.971 | 0.942 | **0.999** | 0.928 | 0.838 | 0.967 | 0.932 | 0.22 | 0.963 | 0.971 | **0.982** | 0.922 | 0.971 | 0.964 |
| Pima | 0.72 | 0.711 | **0.741** | 0.587 | 0.593 | 0.704 | 0.584 | 0.665 | **0.763** | 0.697 | 0.662 | 0.688 | 0.663 | 0.654 |
| Satellite | 0.807 | 0.662 | 0.874 | 0.582 | 0.8 | 0.734 | 0.864 | 0.795 | 0.859 | 0.854 | 0.789 | 0.74 | **0.902** | **0.877** |
| Satimage-2 | 0.993 | 0.979 | **0.999** | 0.965 | 0.961 | 0.998 | 0.997 | 0.994 | 0.861 | 0.997 | 0.988 | 0.998 | 1 | **0.999** |
| Seismic | 0.692 | 0.692 | **0.738** | 0.692 | 0.713 | 0.727 | 0.714 | 0.717 | 0.681 | 0.719 | 0.714 | 0.724 | 0.712 | **0.746** |
| Shuttle | 0.996 | 0.994 | 0.999 | 0.993 | 0.997 | 0.996 | 0.998 | 0.99 | 0.997 | 0.999 | 0.998 | 0.992 | 1 | 1 |
| Smtp (KDDCUP99) | 0.905 | 0.809 | **0.936** | 0.88 | 0.78 | 0.845 | 0.926 | 0.911 | 0.89 | 0.885 | **0.953** | 0.894 | 0.927 | 0.929 |
| Speech | 0.478 | 0.471 | 0.486 | 0.471 | 0.563 | 0.472 | 0.511 | 0.55 | **0.609** | **0.582** | 0.511 | 0.531 | 0.47 | 0.47 |
| Thyroid | **0.989** | 0.984 | 0.976 | 0.978 | 0.869 | 0.969 | 0.948 | 0.689 | 0.886 | 0.987 | **0.99** | 0.904 | 0.975 | 0.983 |
| Vertebral | 0.446 | 0.494 | 0.406 | 0.474 | 0.478 | 0.478 | 0.483 | 0.516 | 0.545 | 0.543 | **0.57** | 0.247 | **0.565** | 0.408 |
| Vowels | 0.779 | 0.644 | 0.975 | 0.597 | 0.895 | 0.891 | 0.969 | 0.925 | **0.988** | 0.979 | **0.983** | 0.753 | 0.982 | 0.938 |
| WBC | 0.947 | 0.949 | 0.947 | 0.907 | 0.904 | 0.946 | 0.928 | 0.663 | 0.765 | 0.908 | 0.913 | 0.77 | **0.964** | **0.952** |
| Wine | 0.93 | 0.927 | 0.952 | 0.729 | 0.854 | 0.899 | **0.964** | 0.961 | 0.96 | 0.944 | **0.974** | 0.835 | 0.909 | 0.851 |
| Yeast | **0.863** | **0.85** | 0.802 | 0.787 | 0.706 | 0.816 | 0.416 | 0.235 | 0.629 | 0.713 | 0.781 | 0.699 | 0.744 | 0.73 |
| Average | 0.847 | 0.826 | **0.879** | 0.79 | 0.818 | 0.848 | 0.841 | 0.745 | 0.855 | 0.877 | **0.879** | 0.766 | **0.884** | 0.865 |

Table 10: Detailed AUC-ROC scores for all methods over 30 datasets in ODDS. This table complements the results shown in Section 3.2. The scores are averaged over 5 random dataset splits. The highest number and second highest number in each row are highlighted in red and blue, respectively. AnoLLM with SmolLM-135M backbone outperforms all other methods in terms of averaged performance.

| Datasets \Methods | Classical Methods | | | | Deep-learning based methods | | | | | | | | AnoLLM | |
|---|---|---|---|---|---|---|---|---|---|---|---|---|---|---|
| | Iforest | PCA | KNN | ECOD | DeepSVDD | RCA | SLAD | GOAD | NeuTral | ICL | DTE | REPEN | 135M | 360M |
| Annthyroid | 0.003 | 0.013 | 0.002 | 0.001 | 0.01 | 0.001 | 0.005 | 0.009 | 0.009 | 0.002 | 0.001 | 0.011 | 0.001 | 0.001 |
| Arrhythmia | 0.002 | 0.004 | 0.004 | 0.004 | 0.004 | 0.004 | 0.005 | 0.035 | 0.008 | 0.008 | 0.009 | 0.016 | 0.004 | 0.004 |
| BreastW | 0.001 | 0.001 | 0 | 0 | 0.003 | 0 | 0.002 | 0 | 0.002 | 0.001 | 0.001 | 0.008 | 0 | 0.001 |
| Cardio | 0.004 | 0 | 0.004 | 0 | 0.012 | 0.001 | 0.003 | 0.018 | 0.013 | 0.015 | 0.005 | 0.02 | 0.004 | 0.004 |
| Ecoli | 0.009 | 0.004 | 0.001 | 0.004 | 0.021 | 0.003 | 0.002 | 0.003 | 0.004 | 0.004 | 0.003 | 0.015 | 0.01 | 0.014 |
| ForestCover | 0.006 | 0 | 0 | 0 | 0.025 | 0.001 | 0.027 | 0.017 | 0.018 | 0.004 | 0.002 | 0.007 | 0.021 | 0.008 |
| Glass | 0.007 | 0.005 | 0.008 | 0.009 | 0.017 | 0.008 | 0.013 | 0.021 | 0.011 | 0.016 | 0.006 | 0.019 | 0.014 | 0.014 |
| Heart | 0.009 | 0.006 | 0.007 | 0.006 | 0.008 | 0.006 | 0.009 | 0.007 | 0.01 | 0.034 | 0.01 | 0.042 | 0.006 | 0.009 |
| Http (KDDCUP99) | 0 | 0 | 0 | 0 | 0.005 | 0 | 0 | 0 | 0.01 | 0 | 0 | 0 | 0 | 0 |
| Ionosphere | 0.008 | 0.005 | 0.006 | 0.005 | 0.005 | 0.004 | 0.002 | 0.003 | 0.001 | 0.001 | 0.003 | 0.037 | 0.005 | 0.006 |
| Letter Recognition | 0.003 | 0.001 | 0.003 | 0.002 | 0.006 | 0.003 | 0.003 | 0.006 | 0.003 | 0.004 | 0.003 | 0.018 | 0.003 | 0.003 |
| Lymphography | 0.037 | 0.007 | 0.015 | 0.013 | 0.01 | 0.008 | 0.01 | 0.014 | 0.015 | 0.018 | 0.006 | 0.058 | 0.008 | 0.002 |
| Mammography | 0.002 | 0.001 | 0.001 | 0 | 0.011 | 0.001 | 0.005 | 0.006 | 0.021 | 0.009 | 0.004 | 0.006 | 0.002 | 0.001 |
| Mulcross | 0 | 0 | 0 | 0 | 0 | 0 | 0.004 | 0 | 0.021 | 0 | 0 | 0.007 | 0 | 0 |
| Musk | 0.005 | 0 | 0 | 0 | 0 | 0 | 0 | 0 | 0 | 0 | 0 | 0.056 | 0 | 0 |
| Optdigits | 0.013 | 0.002 | 0.001 | 0.002 | 0.048 | 0.004 | 0.04 | 0.018 | 0.002 | 0.007 | 0.013 | 0.052 | 0.001 | 0.005 |
| Pendigits | 0.003 | 0 | 0 | 0 | 0.03 | 0.002 | 0.006 | 0.013 | 0.002 | 0.008 | 0 | 0.009 | 0.003 | 0.004 |
| Pima | 0.006 | 0.004 | 0.003 | 0.007 | 0.008 | 0.002 | 0.012 | 0.017 | 0.005 | 0.009 | 0.004 | 0.005 | 0.007 | 0.006 |
| Satellite | 0.007 | 0.001 | 0.001 | 0.001 | 0.008 | 0.001 | 0.001 | 0.004 | 0.002 | 0.002 | 0.003 | 0.004 | 0 | 0 |
| Satimage-2 | 0 | 0 | 0 | 0 | 0.004 | 0 | 0 | 0 | 0.009 | 0 | 0 | 0 | 0 | 0 |
| Seismic | 0.005 | 0.005 | 0.003 | 0.004 | 0.003 | 0.003 | 0.004 | 0.005 | 0.003 | 0.004 | 0.003 | 0.008 | 0.003 | 0.002 |
| Shuttle | 0 | 0 | 0 | 0 | 0 | 0 | 0 | 0 | 0.001 | 0 | 0 | 0.003 | 0 | 0 |
| Smtp (KDDCUP99) | 0.003 | 0 | 0.002 | 0 | 0.021 | 0.002 | 0.002 | 0.002 | 0.017 | 0.002 | 0 | 0.001 | 0.003 | 0.001 |
| Speech | 0.007 | 0.003 | 0.004 | 0.003 | 0.009 | 0.003 | 0.016 | 0.011 | 0.012 | 0.008 | 0.005 | 0.012 | 0.003 | 0.004 |
| Thyroid | 0.001 | 0 | 0 | 0 | 0.025 | 0.001 | 0.004 | 0.021 | 0.005 | 0 | 0.001 | 0.016 | 0.001 | 0.001 |
| Vertebral | 0.019 | 0.031 | 0.019 | 0.009 | 0.033 | 0.018 | 0.021 | 0.032 | 0.023 | 0.016 | 0.028 | 0.033 | 0.038 | 0.024 |
| Vowels | 0.011 | 0.005 | 0.002 | 0.003 | 0.008 | 0.004 | 0.001 | 0.004 | 0.001 | 0.006 | 0.001 | 0.025 | 0.004 | 0.006 |
| WBC | 0.003 | 0.002 | 0.001 | 0.003 | 0.034 | 0.002 | 0.005 | 0.067 | 0.02 | 0.003 | 0.006 | 0.017 | 0.004 | 0.004 |
| Wine | 0.008 | 0.008 | 0.008 | 0.007 | 0.032 | 0.006 | 0.006 | 0.02 | 0.007 | 0.007 | 0.003 | 0.058 | 0.047 | 0.016 |
| Yeast | 0.004 | 0.006 | 0.005 | 0.003 | 0.011 | 0.004 | 0.046 | 0.03 | 0.019 | 0.013 | 0.021 | 0.019 | 0.019 | 0.01 |
| Average | 0.002 | 0.001 | 0.001 | 0 | 0.003 | 0.001 | 0.004 | 0.004 | 0.003 | 0.002 | 0.002 | 0.004 | 0.002 | 0.001 |

Table 11: Standard errors of AUC-ROC scores for all methods over 30 datasets in ODDS. This table complements the results shown in Section 3.2. The standard errors are computed over 5 random dataset splits.

| Datasets \Methods | Classical Methods | | | | Deep-learning based methods | | | | | | | | AnoLLM | |
|---|---|---|---|---|---|---|---|---|---|---|---|---|---|---|
| | Iforest | PCA | KNN | ECOD | DeepSVDD | RCA | SLAD | GOAD | NeuTral | ICL | DTE | REPEN | 135M | 360M |
| Annthyroid | 0.574 | 0.487 | 0.44 | 0.388 | 0.436 | 0.367 | 0.418 | 0.257 | 0.468 | 0.501 | 0.789 | 0.338 | 0.584 | 0.597 |
| Arrhythmia | 0.612 | 0.542 | 0.554 | 0.591 | 0.533 | 0.542 | 0.536 | 0.503 | 0.515 | 0.533 | 0.521 | 0.452 | 0.612 | 0.6 |
| BreastW | 0.969 | 0.959 | 0.963 | 0.954 | 0.936 | 0.959 | 0.951 | 0.966 | 0.967 | 0.959 | 0.963 | 0.935 | 0.958 | 0.966 |
| Cardio | 0.715 | 0.808 | 0.676 | 0.666 | 0.564 | 0.726 | 0.602 | 0.294 | 0.568 | 0.689 | 0.644 | 0.561 | 0.734 | 0.665 |
| Ecoli | 0.756 | 0.778 | 0.778 | 0.311 | 0.533 | 0.778 | 0.778 | 0.778 | 0.511 | 0.711 | 0.66 | 0.756 | 0.333 | 0.333 |
| ForestCover | 0.109 | 0.158 | 0.745 | 0.238 | 0.035 | 0.189 | 0.136 | 0.001 | 0.426 | 0.769 | 0.778 | 0.064 | 0.256 | 0.213 |
| Glass | 0.156 | 0.133 | 0.178 | 0.156 | 0.244 | 0.156 | 0.178 | 0.133 | 0.422 | 0.289 | 0.133 | 0.067 | 0.178 | 0.178 |
| Heart | 0.917 | 0.922 | 0.906 | 0.893 | 0.906 | 0.908 | 0.913 | 0.91 | 0.904 | 0.91 | 0.914 | 0.876 | 0.912 | 0.907 |
| Http (KDDCUP99) | 0.107 | 0.926 | 0.994 | 0.022 | 0.459 | 0.382 | 0.929 | 0.438 | 0.193 | 0.993 | 0.349 | 0.204 | 0.989 | 0.958 |
| Ionosphere | 0.797 | 0.789 | 0.894 | 0.66 | 0.895 | 0.832 | 0.894 | 0.854 | 0.882 | 0.906 | 0.9 | 0.595 | 0.821 | 0.838 |
| Letter Recognition | 0.176 | 0.136 | 0.434 | 0.146 | 0.404 | 0.29 | 0.548 | 0.404 | 0.636 | 0.722 | 0.588 | 0.164 | 0.734 | 0.486 |
| Lymphography | 0.233 | 0.567 | 0.667 | 0.400 | 0.567 | 0.667 | 0.667 | 0.667 | 0.633 | 0.667 | 0.667 | 0.667 | 0.767 | 0.8 |
| Mammography | 0.413 | 0.474 | 0.409 | 0.535 | 0.443 | 0.358 | 0.138 | 0.287 | 0.135 | 0.298 | 0.364 | 0.294 | 0.551 | 0.428 |
| Mulcross | 0.995 | 1 | 1 | 0.747 | 1 | 0.999 | 0.76 | 1 | 0.852 | 0.996 | 1 | 0.816 | 1 | 1 |
| Musk | 0.616 | 1 | 1 | 0.546 | 1 | 1 | 1 | 1 | 1 | 1 | 1 | 0.15 | 1 | 1 |
| Optdigits | 0.159 | 0.001 | 0.284 | 0.027 | 0.297 | 0.021 | 0.04 | 0.165 | 0.639 | 0.471 | 0.164 | 0.02 | 0.72 | 0.443 |
| Pendigits | 0.551 | 0.442 | 0.91 | 0.427 | 0.445 | 0.53 | 0.356 | 0 | 0.467 | 0.61 | 0.606 | 0.373 | 0.559 | 0.505 |
| Pima | 0.672 | 0.688 | 0.692 | 0.578 | 0.57 | 0.672 | 0.585 | 0.628 | 0.695 | 0.67 | 0.624 | 0.668 | 0.626 | 0.62 |
| Satellite | 0.696 | 0.614 | 0.762 | 0.538 | 0.71 | 0.685 | 0.76 | 0.693 | 0.751 | 0.757 | 0.723 | 0.691 | 0.798 | 0.774 |
| Satimage-2 | 0.873 | 0.848 | 0.935 | 0.718 | 0.884 | 0.949 | 0.825 | 0.955 | 0.051 | 0.918 | 0.501 | 0.916 | 0.952 | 0.944 |
| Seismic | 0.251 | 0.266 | 0.291 | 0.282 | 0.258 | 0.32 | 0.285 | 0.295 | 0.195 | 0.299 | 0.239 | 0.306 | 0.279 | 0.316 |
| Shuttle | 0.964 | 0.958 | 0.977 | 0.917 | 0.983 | 0.969 | 0.975 | 0.965 | 0.982 | 0.983 | 0.974 | 0.936 | 0.983 | 0.984 |
| Smtp (KDDCUP99) | 0 | 0.667 | 0.667 | 0.667 | 0.093 | 0.653 | 0.667 | 0.667 | 0.607 | 0.493 | 0.667 | 0.607 | 0.667 | 0.667 |
| Speech | 0.03 | 0.049 | 0.056 | 0.049 | 0.036 | 0.049 | 0.039 | 0.039 | 0.049 | 0.075 | 0.052 | 0.013 | 0.066 | 0.062 |
| Thyroid | 0.789 | 0.723 | 0.643 | 0.626 | 0.557 | 0.6 | 0.639 | 0.417 | 0.35 | 0.778 | 0.804 | 0.346 | 0.682 | 0.725 |
| Vertebral | 0.187 | 0.207 | 0.14 | 0.213 | 0.28 | 0.16 | 0.16 | 0.3 | 0.327 | 0.193 | 0.3 | 0.013 | 0.287 | 0.18 |
| Vowels | 0.248 | 0.2 | 0.684 | 0.22 | 0.548 | 0.424 | 0.716 | 0.46 | 0.768 | 0.74 | 0.78 | 0.244 | 0.76 | 0.556 |
| WBC | 0.819 | 0.79 | 0.714 | 0.562 | 0.705 | 0.724 | 0.667 | 0.381 | 0.181 | 0.733 | 0.619 | 0.219 | 0.79 | 0.762 |
| Wine | 0.7 | 0.62 | 0.7 | 0.38 | 0.42 | 0.6 | 0.72 | 0.76 | 0.7 | 0.62 | 0.72 | 0.44 | 0.5 | 0.52 |
| Yeast | 0.486 | 0.421 | 0.337 | 0.335 | 0.31 | 0.352 | 0.084 | 0.01 | 0.181 | 0.331 | 0.328 | 0.158 | 0.32 | 0.31 |
| Average | 0.519 | 0.572 | 0.648 | 0.460 | 0.535 | 0.562 | 0.566 | 0.508 | 0.535 | 0.651 | 0.612 | 0.430 | 0.647 | 0.611 |

Table 12: Detailed F1 scores for all methods over 30 datasets in ODDS. This table complements the results shown in Section 3.2. The scores are averaged over 5 random dataset splits. The highest number and second highest number in each row are highlighted in red and blue, respectively.

| Datasets \ Methods | Classical Methods | | | | Deep-learning based methods | | | | | | | | AnoLLM | |
|---|---|---|---|---|---|---|---|---|---|---|---|---|---|---|
| | Iforest | PCA | KNN | ECOD | DeepSVDD | RCA | SLAD | GOAD | NeuTral | ICL | DTE | REPEN | 135M | 360M |
| Annthyroid | 0.01 | 0.013 | 0.002 | 0.001 | 0.005 | 0.003 | 0.006 | 0.008 | 0.015 | 0.006 | 0.007 | 0.011 | 0.003 | 0.002 |
| Arrhythmia | 0.009 | 0.003 | 0.013 | 0.009 | 0.007 | 0.007 | 0.01 | 0.044 | 0.009 | 0.017 | 0.017 | 0.007 | 0.003 | 0.003 |
| BreastW | 0.002 | 0.003 | 0.002 | 0.002 | 0.004 | 0.001 | 0.004 | 0.002 | 0.003 | 0.004 | 0.003 | 0.013 | 0.003 | 0.003 |
| Cardio | 0.016 | 0.002 | 0.011 | 0.005 | 0.015 | 0.008 | 0.008 | 0.006 | 0.028 | 0.017 | 0.009 | 0.021 | 0.004 | 0.004 |
| Ecoli | 0.02 | 0 | 0 | 0.02 | 0.08 | 0 | 0 | 0 | 0.051 | 0.024 | 0 | 0.02 | 0.031 | 0.051 |
| ForestCover | 0.003 | 0.001 | 0.002 | 0.001 | 0.01 | 0.003 | 0.033 | 0 | 0.027 | 0.024 | 0 | 0.007 | 0.08 | 0.008 |
| Glass | 0.024 | 0.02 | 0.024 | 0.024 | 0.037 | 0.024 | 0.024 | 0.02 | 0.058 | 0.074 | 0.02 | 0.04 | 0.051 | 0.051 |
| Heart | 0.001 | 0.002 | 0.004 | 0.002 | 0.005 | 0.003 | 0.002 | 0.004 | 0.004 | 0.005 | 0.002 | 0.003 | 0.002 | 0.005 |
| Http (KDDCUP99) | 0.053 | 0.001 | 0 | 0 | 0.12 | 0.017 | 0.001 | 0.022 | 0.096 | 0.001 | 0.009 | 0.013 | 0.002 | 0.013 |
| Ionosphere | 0.009 | 0.007 | 0.011 | 0.003 | 0.009 | 0.008 | 0.002 | 0.01 | 0.001 | 0.003 | 0.002 | 0.02 | 0.006 | 0.011 |
| Letter Recognition | 0.009 | 0.004 | 0.004 | 0.002 | 0.007 | 0.006 | 0.011 | 0.004 | 0.009 | 0.016 | 0.022 | 0.018 | 0.017 | 0.007 |
| Lymphography | 0.03 | 0.056 | 0 | 0.056 | 0 | 0.03 | 0.03 | 0.036 | 0 | 0.03 | 0 | 0.101 | 0.06 | 0.03 |
| Mammography | 0.012 | 0.005 | 0.004 | 0.002 | 0.027 | 0.004 | 0.006 | 0.002 | 0.007 | 0.008 | 0.011 | 0.009 | 0.01 | 0.006 |
| Mulcross | 0.001 | 0 | 0 | 0 | 0 | 0 | 0.028 | 0 | 0.077 | 0 | 0 | 0.05 | 0 | 0 |
| Musk | 0.055 | 0 | 0 | 0 | 0 | 0 | 0 | 0 | 0 | 0 | 0 | 0.021 | 0 | 0 |
| Optdigits | 0.023 | 0.001 | 0.012 | 0 | 0.071 | 0.001 | 0.031 | 0.04 | 0.029 | 0.048 | 0.021 | 0.012 | 0.006 | 0.024 |
| Pendigits | 0.02 | 0.005 | 0.004 | 0.002 | 0.062 | 0.013 | 0.024 | 0 | 0.024 | 0.045 | 0.009 | 0.018 | 0.026 | 0.024 |
| Pima | 0.006 | 0.004 | 0.007 | 0.005 | 0.006 | 0.004 | 0.011 | 0.012 | 0.004 | 0.008 | 0.004 | 0.003 | 0.01 | 0.006 |
| Satellite | 0.003 | 0.002 | 0.001 | 0 | 0.008 | 0 | 0.002 | 0.002 | 0.003 | 0.001 | 0 | 0.003 | 0.001 | 0.002 |
| Satimage-2 | 0.007 | 0.006 | 0.008 | 0 | 0.01 | 0.005 | 0.016 | 0.005 | 0.003 | 0.003 | 0.016 | 0.009 | 0.006 | 0.011 |
| Seismic | 0.013 | 0.004 | 0.01 | 0.003 | 0.008 | 0.008 | 0.011 | 0.006 | 0.007 | 0.007 | 0.01 | 0.008 | 0.008 | 0.003 |
| Shuttle | 0.005 | 0.001 | 0.001 | 0 | 0.001 | 0 | 0.001 | 0.001 | 0.001 | 0.001 | 0.001 | 0.026 | 0 | 0 |
| Smtp (KDDCUP99) | 0 | 0 | 0 | 0 | 0.056 | 0.007 | 0 | 0 | 0.015 | 0.034 | 0 | 0.04 | 0 | 0 |
| Speech | 0.003 | 0 | 0.004 | 0 | 0.01 | 0 | 0.007 | 0.011 | 0.011 | 0.01 | 0.005 | 0.006 | 0 | 0.003 |
| Thyroid | 0.016 | 0.006 | 0.004 | 0.01 | 0.048 | 0.008 | 0.023 | 0.01 | 0.013 | 0.007 | 0.006 | 0.017 | 0.013 | 0.018 |
| Vertebral | 0.022 | 0.024 | 0.022 | 0.007 | 0.041 | 0.015 | 0.026 | 0.016 | 0.026 | 0.024 | 0.025 | 0.012 | 0.035 | 0.029 |
| Vowels | 0.017 | 0.01 | 0.017 | 0 | 0.017 | 0.013 | 0.018 | 0.008 | 0.02 | 0.036 | 0.011 | 0.04 | 0.045 | 0.017 |
| WBC | 0.016 | 0.017 | 0 | 0.016 | 0.075 | 0.008 | 0.019 | 0.073 | 0.041 | 0.01 | 0.013 | 0.032 | 0.022 | 0.025 |
| Wine | 0.04 | 0.044 | 0.04 | 0.052 | 0.087 | 0.057 | 0.044 | 0.078 | 0.04 | 0.044 | 0.018 | 0.112 | 0.028 | 0.052 |
| Yeast | 0.013 | 0.014 | 0.011 | 0.002 | 0.028 | 0.013 | 0.015 | 0 | 0.021 | 0.009 | 0.038 | 0.021 | 0.025 | 0.014 |
| Average | 0.003 | 0.002 | 0.002 | 0.002 | 0.001 | 0.003 | 0.003 | 0.002 | 0.005 | 0.004 | 0.003 | 0.007 | 0.006 | 0.004 |

Table 13: Standard error of F1 scores for all methods over 30 datasets in ODDS. This table complements the results shown in Section 3.2. The standard errors are computed over 5 random dataset splits.

| Datasets \ Methods | Classical Methods | | | | Deep-learning based methods | | | | | | | | AnoLLM | |
|---|---|---|---|---|---|---|---|---|---|---|---|---|---|---|
| | Iforest | PCA | KNN | ECOD | DeepSVDD | RCA | SLAD | GOAD | NeuTral | ICL | DTE | REPEN | 135M | 360M |
| Annthyroid | 0.646 | 0.55 | 0.463 | 0.406 | 0.441 | 0.383 | 0.461 | 0.286 | 0.435 | 0.555 | **0.835** | 0.343 | 0.631 | **0.648** |
| Arrhythmia | **0.662** | 0.617 | 0.556 | 0.622 | 0.563 | 0.562 | 0.556 | 0.518 | 0.507 | 0.556 | 0.566 | 0.419 | 0.636 | **0.642** |
| BreastW | **0.994** | 0.985 | 0.992 | 0.992 | 0.966 | 0.986 | 0.983 | **0.994** | 0.97 | 0.991 | 0.967 | 0.925 | 0.991 | 0.992 |
| Cardio | 0.786 | **0.844** | 0.737 | 0.712 | 0.606 | 0.745 | 0.667 | 0.325 | 0.61 | 0.75 | 0.678 | 0.567 | **0.811** | 0.726 |
| Ecoli | 0.086 | 0.16 | **0.739** | 0.189 | 0.024 | 0.176 | 0.11 | 0.012 | 0.43 | 0.664 | **0.777** | 0.093 | 0.206 | 0.127 |
| ForestCover | 0.649 | 0.71 | 0.786 | 0.306 | 0.621 | 0.739 | 0.728 | **0.796** | 0.475 | **0.807** | 0.583 | 0.682 | 0.419 | 0.417 |
| Glass | 0.198 | 0.167 | 0.242 | 0.242 | 0.263 | 0.187 | 0.208 | 0.15 | **0.484** | **0.374** | 0.226 | 0.165 | 0.247 | 0.234 |
| Heart | 0.972 | **0.976** | 0.972 | 0.94 | 0.964 | 0.966 | 0.973 | **0.977** | 0.972 | 0.963 | 0.975 | 0.884 | 0.972 | 0.969 |
| Http (KDDCUP99) | 0.496 | 0.911 | **0.995** | 0.254 | 0.588 | 0.403 | 0.91 | 0.618 | 0.364 | **0.995** | 0.583 | 0.536 | 0.97 | 0.956 |
| Ionosphere | 0.898 | 0.912 | 0.967 | 0.769 | 0.967 | 0.932 | 0.969 | 0.958 | 0.959 | **0.977** | **0.972** | 0.534 | 0.933 | 0.932 |
| Letter Recognition | 0.168 | 0.143 | 0.426 | 0.141 | 0.412 | 0.259 | 0.578 | 0.39 | 0.703 | **0.773** | 0.55 | 0.151 | **0.797** | 0.191 |
| Lymphography | 0.232 | 0.624 | 0.720 | 0.365 | 0.680 | 0.783 | 0.795 | 0.697 | 0.681 | 0.718 | 0.747 | 0.697 | **0.856** | **0.938** |
| Mammography | 0.392 | 0.443 | 0.399 | **0.548** | 0.447 | 0.312 | 0.126 | 0.232 | 0.094 | 0.287 | 0.378 | 0.268 | **0.592** | 0.364 |
| Mulcross | 0.989 | **1** | **1** | 0.722 | **1** | **1** | 0.788 | **1** | 0.816 | 0.998 | **1** | 0.782 | **1** | **1** |
| Musk | 0.666 | **1** | **1** | 0.627 | **1** | **1** | **1** | **1** | **1** | **1** | **1** | 0.175 | **1** | **1** |
| Optdigits | 0.166 | 0.059 | 0.314 | 0.065 | 0.232 | 0.122 | 0.109 | 0.178 | **0.645** | 0.414 | 0.222 | 0.076 | **0.75** | 0.398 |
| Pendigits | 0.544 | 0.376 | **0.958** | 0.395 | 0.416 | 0.516 | 0.292 | 0.026 | 0.408 | **0.656** | 0.509 | 0.319 | 0.623 | 0.554 |
| Pima | 0.714 | 0.696 | **0.716** | 0.622 | 0.606 | 0.698 | 0.603 | 0.66 | **0.746** | 0.695 | 0.639 | 0.673 | 0.677 | 0.674 |
| Satellite | 0.845 | 0.769 | 0.889 | 0.658 | 0.842 | 0.806 | 0.866 | 0.808 | 0.86 | 0.887 | 0.843 | 0.806 | **0.91** | **0.891** |
| Satimage-2 | 0.93 | 0.901 | **0.98** | 0.745 | 0.905 | 0.977 | 0.903 | **0.98** | 0.082 | 0.967 | 0.526 | 0.952 | **0.988** | 0.974 |
| Seismic | 0.235 | 0.216 | **0.256** | 0.244 | 0.226 | 0.25 | 0.241 | 0.239 | 0.193 | 0.25 | 0.224 | 0.249 | 0.236 | **0.281** |
| Shuttle | 0.984 | 0.962 | 0.972 | 0.946 | 0.987 | 0.96 | 0.968 | 0.949 | 0.994 | 0.995 | 0.946 | 0.928 | **0.997** | **0.996** |
| Smtp (KDDCUP99) | 0.01 | 0.454 | 0.459 | 0.608 | 0.058 | 0.441 | 0.469 | 0.441 | 0.582 | 0.377 | 0.467 | 0.403 | **0.658** | **0.645** |
| Speech | 0.035 | 0.037 | 0.038 | 0.04 | 0.042 | 0.037 | 0.036 | 0.04 | **0.052** | **0.057** | 0.04 | 0.035 | 0.036 | 0.037 |
| Thyroid | 0.783 | 0.791 | 0.696 | 0.635 | 0.56 | 0.649 | 0.686 | 0.401 | 0.33 | **0.822** | **0.86** | 0.385 | 0.696 | 0.74 |
| Vertebral | 0.21 | 0.232 | 0.192 | 0.228 | 0.252 | 0.214 | 0.21 | 0.281 | **0.303** | 0.264 | **0.31** | 0.151 | 0.289 | 0.181 |
| Vowels | 0.229 | 0.162 | 0.762 | 0.153 | 0.603 | 0.455 | 0.765 | 0.544 | **0.861** | 0.804 | 0.831 | 0.203 | **0.839** | 0.599 |
| WBC | 0.842 | **0.876** | 0.814 | 0.586 | 0.747 | 0.808 | 0.711 | 0.408 | 0.226 | 0.714 | 0.64 | 0.235 | **0.873** | 0.753 |
| Wine | 0.672 | 0.659 | 0.711 | 0.321 | 0.512 | 0.517 | 0.782 | **0.789** | 0.779 | 0.734 | **0.873** | 0.484 | 0.522 | 0.529 |
| Yeast | **0.44** | **0.346** | 0.294 | 0.323 | 0.299 | 0.303 | 0.106 | 0.076 | 0.168 | 0.262 | 0.282 | 0.187 | 0.301 | 0.302 |
| Average | 0.549 | 0.586 | 0.668 | 0.480 | 0.561 | 0.573 | 0.587 | 0.526 | 0.558 | **0.677** | 0.635 | 0.444 | **0.682** | 0.623 |

Table 14: Detailed AUC-PR scores for all methods over 30 datasets in ODDS. This table complements the results shown in Section 3.2. The scores are averaged over 5 random dataset splits. The highest number and second highest number in each row are highlighted in red and blue, respectively.

| Datasets \ Methods | Classical Methods | | | | Deep-learning based methods | | | | | | | | AnoLLM | |
|---|---|---|---|---|---|---|---|---|---|---|---|---|---|---|
| | Iforest | PCA | KNN | ECOD | DeepSVDD | RCA | SLAD | GOAD | NeuTral | ICL | DTE | REPEN | 135M | 360M |
| Annthyroid | 0.01 | 0.019 | 0.002 | 0.004 | 0.007 | 0.003 | 0.003 | 0.007 | 0.022 | 0.003 | 0.004 | 0.011 | 0.005 | 0.004 |
| Arrhythmia | 0.011 | 0.01 | 0.013 | 0.014 | 0.011 | 0.014 | 0.011 | 0.031 | 0.015 | 0.022 | 0.012 | 0.01 | 0.015 | 0.013 |
| BreastW | 0.001 | 0.003 | 0 | 0.001 | 0.005 | 0.001 | 0.002 | 0 | 0.006 | 0.002 | 0.004 | 0.016 | 0 | 0.001 |
| Cardio | 0.01 | 0.005 | 0.007 | 0.002 | 0.015 | 0.004 | 0.008 | 0.005 | 0.034 | 0.013 | 0.008 | 0.021 | 0.003 | 0.003 |
| Ecoli | 0.044 | 0.027 | 0.011 | 0.028 | 0.072 | 0.023 | 0.024 | 0.005 | 0.026 | 0.028 | 0.013 | 0.032 | 0.036 | 0.044 |
| ForestCover | 0.004 | 0.001 | 0.005 | 0.001 | 0.003 | 0.002 | 0.03 | 0 | 0.035 | 0.022 | 0.012 | 0.003 | 0.084 | 0.007 |
| Glass | 0.007 | 0.008 | 0.011 | 0.005 | 0.015 | 0.012 | 0.002 | 0.006 | 0.063 | 0.066 | 0.018 | 0.013 | 0.022 | 0.039 |
| Heart | 0.002 | 0.001 | 0.001 | 0.001 | 0.001 | 0.001 | 0.002 | 0.001 | 0.002 | 0.007 | 0.002 | 0.012 | 0.001 | 0.002 |
| Http (KDDCUP99) | 0.023 | 0.001 | 0 | 0 | 0.095 | 0.007 | 0.002 | 0.009 | 0.102 | 0.001 | 0.005 | 0.004 | 0.009 | 0.015 |
| Ionosphere | 0.009 | 0.005 | 0.005 | 0.003 | 0.005 | 0.004 | 0.001 | 0.002 | 0.002 | 0 | 0.001 | 0.027 | 0.003 | 0.006 |
| Letter Recognition | 0.001 | 0.002 | 0.007 | 0.001 | 0.006 | 0.005 | 0.016 | 0.008 | 0.006 | 0.016 | 0.02 | 0.007 | 0.015 | 0.003 |
| Lymphography | 0.016 | 0.074 | 0.01 | 0.018 | 0.008 | 0.019 | 0.018 | 0.021 | 0.007 | 0.023 | 0.001 | 0.105 | 0.036 | 0.016 |
| Mammography | 0.016 | 0.003 | 0.004 | 0.004 | 0.031 | 0.007 | 0.006 | 0.009 | 0.007 | 0.012 | 0.007 | 0.011 | 0.007 | 0.007 |
| Mulcross | 0.001 | 0 | 0 | 0 | 0 | 0 | 0.016 | 0 | 0.086 | 0 | 0 | 0.043 | 0 | 0 |
| Musk | 0.063 | 0 | 0 | 0.005 | 0 | 0 | 0 | 0 | 0 | 0 | 0 | 0.034 | 0 | 0 |
| Optdigits | 0.013 | 0 | 0.004 | 0 | 0.065 | 0.002 | 0.022 | 0.022 | 0.023 | 0.034 | 0.015 | 0.01 | 0.016 | 0.021 |
| Pendigits | 0.029 | 0.003 | 0.006 | 0.003 | 0.069 | 0.015 | 0.013 | 0 | 0.016 | 0.043 | 0.004 | 0.012 | 0.023 | 0.022 |
| Pima | 0.007 | 0.004 | 0.003 | 0.006 | 0.007 | 0.004 | 0.011 | 0.012 | 0.011 | 0.008 | 0.003 | 0.006 | 0.004 | 0.01 |
| Satellite | 0.004 | 0.001 | 0.001 | 0.001 | 0.007 | 0.001 | 0.001 | 0.003 | 0.001 | 0.001 | 0.001 | 0.005 | 0.001 | 0.001 |
| Satimage-2 | 0.004 | 0.001 | 0.002 | 0.004 | 0.006 | 0.001 | 0.015 | 0 | 0.004 | 0.001 | 0.023 | 0.01 | 0.002 | 0.004 |
| Seismic | 0.008 | 0.006 | 0.003 | 0.005 | 0.002 | 0.001 | 0.005 | 0.004 | 0.004 | 0.005 | 0.002 | 0.007 | 0.003 | 0.001 |
| Shuttle | 0.002 | 0.002 | 0.002 | 0.001 | 0.002 | 0.001 | 0.003 | 0.002 | 0.001 | 0.001 | 0.002 | 0.027 | 0 | 0.001 |
| Smtp (KDDCUP99) | 0 | 0.015 | 0.015 | 0.014 | 0.036 | 0.016 | 0.009 | 0.012 | 0.013 | 0.051 | 0.007 | 0.021 | 0.003 | 0.009 |
| Speech | 0.002 | 0 | 0 | 0.001 | 0.001 | 0 | 0.002 | 0.002 | 0.004 | 0.004 | 0.003 | 0.001 | 0 | 0 |
| Thyroid | 0.024 | 0.005 | 0.007 | 0.011 | 0.058 | 0.008 | 0.024 | 0.011 | 0.014 | 0.019 | 0.006 | 0.024 | 0.021 | 0.019 |
| Vertebral | 0.01 | 0.018 | 0.009 | 0.006 | 0.023 | 0.009 | 0.008 | 0.022 | 0.016 | 0.019 | 0.025 | 0.006 | 0.026 | 0.01 |
| Vowels | 0.023 | 0.015 | 0.016 | 0.014 | 0.011 | 0.018 | 0.022 | 0.018 | 0.016 | 0.034 | 0.014 | 0.021 | 0.04 | 0.025 |
| WBC | 0.037 | 0.022 | 0.017 | 0.016 | 0.089 | 0.017 | 0.029 | 0.077 | 0.019 | 0.03 | 0.027 | 0.017 | 0.014 | 0.02 |
| Wine | 0.061 | 0.034 | 0.043 | 0.018 | 0.085 | 0.029 | 0.041 | 0.105 | 0.024 | 0.04 | 0.016 | 0.113 | 0.015 | 0.042 |
| Yeast | 0.009 | 0.01 | 0.006 | 0.005 | 0.026 | 0.007 | 0.011 | 0.003 | 0.011 | 0.013 | 0.026 | 0.01 | 0.021 | 0.013 |
| Average | 0.005 | 0.003 | 0.002 | 0.001 | 0.004 | 0.002 | 0.004 | 0.002 | 0.006 | 0.003 | 0.003 | 0.006 | 0.004 | 0.003 |

Table 15: Standard error of AUC-PR scores for all methods over 30 datasets in ODDS. This table complements the results shown in Section 3.2. The standard errors are computed over 5 random dataset splits.

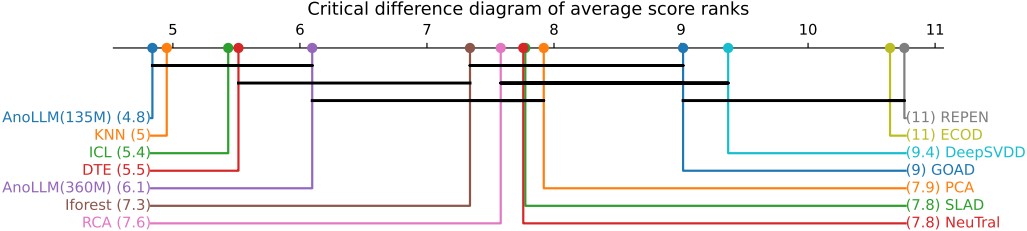

Figure 7: Critical difference diagram of average ranks for various methods on ODDS benchmark. The ranks are computed over AUC-ROC scores. The x-axis shows the average ranks across datasets, with smaller values indicating better performance. Horizontal bars connect groups of methods that are not significantly different in performance according to a statistical test. While AnoLLM(135M) achieves the best average rank (4.8), it is not significantly better than the best-performing baselines, KNN, ICL, and DTE.

| Dataset \ Methods | Equal-width | Quantile | Language | No binning | Standard |
|---|---|---|---|---|---|
| Annthyroid | 0.684 | 0.85 | 0.63 | 0.896 | **0.927** |
| Arrhythmia | 0.827 | **0.832** | 0.824 | 0.814 | 0.825 |
| Breastw | 0.991 | 0.982 | 0.991 | 0.991 | **0.992** |
| Cardio | 0.903 | 0.882 | 0.875 | 0.912 | **0.94** |
| Covertype | **0.937** | 0.442 | 0.936 | 0.643 | 0.881 |
| Ecoli | 0.795 | 0.831 | **0.854** | 0.729 | 0.777 |
| Glass | **0.879** | 0.401 | 0.835 | 0.392 | 0.819 |
| Heart | 0.808 | 0.804 | 0.799 | **0.827** | 0.82 |
| Http (KDDCUP99) | **1.000** | 0.166 | **1.000** | 0.999 | **1.000** |
| Ionosphere | 0.912 | 0.763 | **0.941** | 0.164 | 0.909 |
| Letter Recognition | 0.893 | 0.888 | 0.861 | 0.892 | **0.967** |
| Lymphography | 0.997 | 0.993 | **0.998** | 0.988 | 0.968 |
| Mammography | 0.855 | 0.853 | 0.849 | 0.828 | **0.915** |
| Mulcross | **1.000** | **1.000** | **1.000** | 0.996 | **1.000** |
| Musk | **1.000** | 0.691 | **1.000** | **1.000** | **1.000** |
| Optdigits | 0.912 | 0.691 | 0.939 | 0.942 | **0.983** |
| Pendigits | 0.958 | 0.909 | 0.958 | 0.794 | **0.971** |
| Pima | **0.714** | 0.609 | 0.712 | 0.615 | 0.663 |
| Satellite | 0.843 | 0.892 | 0.842 | 0.888 | **0.902** |
| Satimage-2 | 0.992 | 0.83 | 0.996 | 0.999 | **1.000** |
| Seismic | 0.733 | 0.698 | **0.74** | 0.732 | 0.712 |
| Shuttle | 0.994 | 0.991 | 0.987 | **1.000** | **1.000** |
| Smtp(KDDCUP99) | 0.89 | 0.501 | 0.816 | 0.909 | **0.927** |
| Speech | 0.456 | 0.479 | 0.46 | **0.545** | 0.47 |
| Thyroid | 0.951 | 0.812 | 0.963 | 0.97 | **0.975** |
| Vertebral | 0.461 | 0.54 | 0.461 | 0.471 | **0.565** |
| Vowels | 0.946 | 0.855 | 0.957 | 0.555 | **0.982** |
| WBC | 0.938 | 0.84 | **0.969** | 0.857 | 0.964 |
| Wine | 0.885 | 0.845 | 0.895 | 0.907 | **0.909** |
| Yeast | 0.794 | 0.697 | **0.804** | 0.744 | 0.744 |
| Average | 0.865 | 0.752 | 0.863 | 0.800 | **0.884** |

Table 16: Comparison of different binning methods. The table shows AUC-ROC scores of different binning methods for all datasets in ODDS library. This table complements the results shown in Section 3.3. As can be seen from the table, AnoLLM with standard rescaling performs the best and achieve the best performance on 18 out of 30 datasets from ODDS.

| Mixed-type benchmark | | | |
| --- | --- | --- | --- |
| Datasets \ LLM backbone | SmolLM-135M | SmolLM-360M | SmolLM-1.7B |
| fake job post | 0.800 | **0.814** | 0.802 |
| fraud ecommerce | **1** | 0.999 | 0.999 |
| Lymphography | 0.968 | **0.995** | **0.995** |
| Seismic | 0.712 | **0.746** | 0.74 |
| Vehicle insurance | **0.569** | 0.557 | 0.56 |
| 20 newsgroups | 0.766 | 0.752 | 0.774 |
| Average (Mixed-type) | 0.803 | 0.811 | **0.812** |
| ODDS benchmark | | | |
| Datasets \ LLM backbone | SmolLM-135M | SmolLM-360M | SmolLM-1.7B |
| Annthyroid | 0.927 | **0.931** | 0.93 |
| Arrhythmia | **0.825** | 0.822 | 0.824 |
| BreastW | 0.992 | **0.993** | 0.991 |
| Cardio | **0.94** | 0.873 | 0.867 |
| Ecoli | 0.777 | **0.804** | 0.791 |
| ForestCover | 0.881 | 0.835 | **0.887** |
| Glass | **0.819** | 0.797 | 0.818 |
| Heart | **0.82** | 0.799 | 0.803 |
| Http (KDDCUP99) | **1** | **1** | **1** |
| Ionosphere | 0.909 | **0.924** | 0.918 |
| Letter Recognition | **0.967** | 0.867 | 0.772 |
| Lymphography | 0.968 | 0.993 | **0.995** |
| Mammography | **0.915** | 0.876 | 0.874 |
| Mulcross | **1** | **1** | **1** |
| Musk | **1** | **1** | **1** |
| Optdigits | **0.983** | 0.939 | 0.888 |
| Pendigits | **0.971** | 0.964 | 0.93 |
| Pima | **0.663** | 0.654 | 0.653 |
| Satellite | **0.902** | 0.877 | 0.858 |
| Satimage-2 | **1** | 0.999 | 0.999 |
| Seismic | 0.712 | **0.746** | 0.74 |
| Shuttle | **1** | **1** | 0.999 |
| Smtp (KDDCUP99) | 0.927 | **0.929** | 0.924 |
| Speech | **0.47** | **0.47** | **0.47** |
| Thyroid | 0.975 | 0.983 | **0.991** |
| Vertebral | **0.565** | 0.408 | 0.392 |
| Vowels | **0.982** | 0.938 | 0.933 |
| WBC | **0.964** | 0.952 | 0.957 |
| Wine | **0.909** | 0.851 | 0.876 |
| Yeast | 0.744 | 0.73 | **0.754** |
| Average (ODDS) | **0.884** | 0.865 | 0.861 |

Table 17: Comparison of different LLM sizes in the AnoLLM framework. The table shows AUC-ROC scores of different backbone LLMs of the AnoLLM from the mixed-type benchmark and ODDS library. This table complements the results shown in Section 3.4.

