# OpenReview forum: "AnoLLM: Large Language Models for Tabular Anomaly Detection"
_ICLR.cc/2025/Conference — ICLR 2025 Poster_

### Official Review · Reviewer_Sz6K · 2024-11-01

**Soundness:** 2
**Presentation:** 2
**Contribution:** 2
**Rating:** 5
**Confidence:** 4

**Summary:**

This paper proposes AnoLLM, a new framework that uses large language models (LLMs) for unsupervised tabular anomaly detection. AnoLLM assigns anomaly scores based on the negative log likelihood for anomaly detection. The authors claim that AnoLLM detects anomal samples in raw features and can deal with textual and categorical features. Experimental results show that AnoLLM achieves good performance on six benchmark datasets with mixed feature types. AnoLLM is also competitive with KNN.

**Strengths:**

It’s good to consider different types in tabular data, such as textual, numerical and categorical columns.

Experiment results show good potential for AnoLLM.

**Weaknesses:**

The method seems too simple and lack novelty. It’s easy and direct to consider to use the negative log likelihood for anomaly detection. Even though authors consider different types of columns in tabular data, the process seems to be the same.

I think it’s inappropriate to claim that you are the first to apply LLMs to tabular anomaly detection. There are many works in this area, such as “Anomaly detection of tabular data using llms”, and other corresponding works such as “LLMClean: Context-Aware Tabular Data Cleaning via LLM-Generated OFDs”, “Enhancing Anomaly Detection in Financial Markets with an LLM-based Multi-Agent Framework”.

The backbone LLM is not so famous such as Llama, Qwen, Mistral, etc. Why authors do not use these LLMs?

I don’t understand what the differences in contribution 3 and 4. It seems they are all talking about the experiment.

In experiments, only one of the datasets here has text columns. More such datasets and the dataset with more attributes (features) should be considered.

More ablation studies should be provided such as the impact of Random column permutations. Case study is also missing, I cannot recognize which samples are anomaly.

By the way, the formatting on line 249 could use some improvement.

**Questions:**

See weakness.

---

> ### Author Response · Authors · 2024-11-22
> **Thank you for your review.**
>
> Thank you for your review. We sincerely appreciate the time you took to read our paper and are grateful for your feedback. Our responses are provided below.
>
> **Regarding the simplicity of proposed methods:**  We consider simplicity a key advantage of our method, as it makes implementation more accessible for practitioners and facilitates easier debugging. A major contribution of our work is demonstrating that LLMs can effectively handle tabular data, despite its sequential structure and the challenges of numerical reasoning. By employing techniques such as random permutation and number normalization, LLMs can be adapted for tabular anomaly detection while leveraging their strengths in text modeling. We view this as a solid starting point, leaving additional modifications and exploration of more advanced techniques to future research.
>
> **Addressing claims of pioneering the use of LLMs for tabular anomaly detection:** We would like to highlight that AnoLLM fundamentally differs from the works you mentioned. Biester et al. (2024) and Park (2024) treat LLMs as agents for generating domain-specific contexts or formatting data, relying on additional modules—and in some cases, human intervention—to process the LLMs’ outputs. Li et al. (2024), on the other hand, focuses on zero-shot performance. In contrast, our approach directly fine-tunes LLMs on the target data and uses the LLMs’ outputs as anomaly scores, making it a more straightforward and self-contained method.
>
> **Choice of LLM backbones:**  We experimented with Qwen, a larger architecture with 500M parameters in its smallest variant, and other multi-billion scale models. However, we found this task was better suited to lower-capacity models fine-tuned (FT) for anomaly detection (AD), as AD typically requires high throughput and low latency, making high-capacity, multi-billion parameter models impractical. As shown in Table 4, scaling the size of LLMs did not yield performance improvements. To optimize efficiency, we focused on smaller LLMs as backbones. SmolLM, the state-of-the-art open-weight model at the time, was selected for our experiments, and initial trials with Qwen-0.5B showed comparable performance, reinforcing our preference for smaller models.
>
> **Distinction between Contributions 3 and 4:** Contributions 3 and 4 focus on different aspects of experimental results across various datasets. Contribution 3 highlights AnoLLM's strength with datasets containing mixed-type features, where it consistently outperforms all other methods. In contrast, Contribution 4 addresses the ODDS benchmark, which is dominated by numerical features (over 98.5%). Despite this, AnoLLMs perform comparably to the best methods. We distinguish these two contributions to clearly highlight this difference.
>
> **Consideration of additional datasets containing textual features:** We note that there are two datasets containing textual features, fake job posts and 20 newsgroups. Although we have explored other tabular datasets with textual features, they either lack a legal license for our use or are not publicly available. Additionally, we would like to emphasize that AnoLLM performs well on tabular datasets with mixed-type attributes, including both categorical and numerical features. We identify a total of six datasets that meet this criterion.
>
>
> **Impact of random column permutations:**  We conducted an ablation study on the effect of random permutations, detailed in Section C. The results indicate that random permutation is a critical component of AnoLLM, and its absence can lead to a significant decline in performance.
>
> **Case study of AnoLLM:** One failure case of AnoLLM is that the negative log-likelihood assigns equal importance to all features, which can be problematic when certain features are more critical than others. For instance, in the wine dataset, we observed that a single feature, Proline, plays a key role in distinguishing anomalies. AnoLLM performs worse on this dataset because it aggregates anomaly scores across all features, diluting the influence of Proline. In contrast, methods like KNN can better identify the importance of Proline, as its significantly larger values dominate the anomaly scores, leading to more accurate predictions.
>
> **Formatting of line 249 (Eqn.)** Thanks for your suggestion. We have reformatted the Eqn.6.

---

> > ### Comment · Reviewer_Sz6K · 2024-11-27
> >
> > Thanks to the authors for the reply. It does address some of my questions. But I still think it’s inappropriate to claim that you are the first to apply LLMs to tabular anomaly detection, which can not be the contribution. Experiments with different LLM backbones can not be found.
> >
> > I also read the other reviewers' comments carefully and I will review my score during the discussion phase.

---

> > > ### Author Response · Authors · 2024-12-01
> > > **Thank you for your responses.**
> > >
> > > Thank you for your responses. We are pleased that some of your questions have been resolved. We hope the following clarifications address your remaining concerns:
> > >
> > > **Claim of pioneering the use of LLMs for tabular anomaly detection:** We apologize for not explicitly mentioning earlier that we have removed this claim in the last sentence of the abstract in the updated draft. The works you referenced are also discussed in the Related Work section.
> > >
> > > **Experiments with other LLM backbones:** We apologize for not being able to run a full set of experiments comparing different LLM backbones due to limited computational resources. However, we performed a comparison between Qwen-500M and SmolLM-135M using 24 smaller datasets from ODDS. The results indicate that, while there are minor variations across individual datasets, the overall average performance remains comparable. Consequently, we decided to proceed with the smaller LLM backbone.
> > >
> > >
> > > | Dataset \\ Model   | Qwen2-500M | SmolLM-135M |
> > > | ------------------ | --------- | ----------- |
> > > | Annthyroid         | 0.984     | 0.927       |
> > > | BreastW            | 0.99      | 0.992       |
> > > | Cardio             | 0.941     | 0.94        |
> > > | Ecoli              | 0.842     | 0.777       |
> > > | ForestCover        | 0.999     | 0.881       |
> > > | Glass              | 0.856     | 0.819       |
> > > | Heart              | 0.854     | 0.82        |
> > > | Ionosphere         | 0.928     | 0.909       |
> > > | Letter Recognition | 0.98      | 0.967       |
> > > | Lymphography       | 0.987     | 0.968       |
> > > | Mammography        | 0.864     | 0.915       |
> > > | Optdigits          | 0.989     | 0.983       |
> > > | Pendigits          | 0.935     | 0.971       |
> > > | Pima               | 0.696     | 0.663       |
> > > | Satellite          | 0.864     | 0.902       |
> > > | Satimage-2         | 0.989     | 1           |
> > > | Shuttle            | 0.994     | 1           |
> > > | Smtp (KDDCUP99)    | 0.896     | 0.927       |
> > > | Thyroid            | 0.953     | 0.975       |
> > > | Vertebral          | 0.643     | 0.565       |
> > > | Vowels             | 0.997     | 0.982       |
> > > | WBC                | 0.951     | 0.964       |
> > > | Wine               | 0.904     | 0.909       |
> > > | Yeast              | 0.748     | 0.744       |
> > > | Average            | 0.908     | 0.896       |

---

### Official Review · Reviewer_Eu7r · 2024-11-03

**Soundness:** 3
**Presentation:** 3
**Contribution:** 3
**Rating:** 8
**Confidence:** 4

**Summary:**

This paper presents a new framework, AnoLLM, for unsupervised anomaly detection by fine-tuning a pretrained large language model (LLM). The authors use a predefined template to serialize, i.e., convert tabular data into text for the LLM, along with preprocessing to mitigate limitations related to the model's autoregressive nature. They employ the negative log-likelihood across different column permutations to compute an anomaly score for each sample in the test set. The method is compared against various classical and deep learning methods on the ODDS datasets and six new datasets featuring mixed types of attributes. Overall, the approach demonstrates strong performance against baselines, particularly for datasets containing text features.

**Strengths:**

- The paper introduces a novel model type for anomaly detection using a large language model.
- The method provides an effective way to handle text and categorical data as features for anomaly detection, which are typically challenging to manage. To do so, they proposed a method to mitigate length-bias in the LLM’s output probabilities and theoretically validated it.
- The authors demonstrate a preprocessing technique for tabular data, facilitating effective LLM fine-tuning.
- The work is easy to follow and the motivation is clear.

**Weaknesses:**

The paper claims to outperform certain deep learning methods; however, in my experience, some of these methods perform similarly or even better than KNN (which is reported to have results comparable to the proposed method). For example, ICL outperforms KNN on the ODDS benchmark (Shenkar and Wolf, 2022), as does DTE (Livernoche et al., 2024), which was cited but not included as a baseline. The use of column permutations in the paper can be seen as a sort of ensemble strategy, a technique known to slightly improve anomaly detection performance. To ensure a fair comparison, the baselines should also be evaluated using these same permutations, as Appendix C suggests that this step may not be critical, or specific, to AnoLLM. In a small test I conducted, implementing this strategy led to performance improvements in other deep learning methods as well. Including F1-score or AUC-PR results as supplemental material would be helpful, as these metrics are more sensitive to class imbalances, which are common in anomaly detection. Scoring metrics can influence the relative ranking of methods on benchmarks. This claim that AnoLLM outperform deep learning methods should be more cautiously framed.

One key limitation mentioned at the end of the paper is the computational expensiveness of the proposed method. 7 A100 GPUs were used for LLM fine-tuning, this makes it difficult for others to access this model or replicate results of the paper. Since no code is provided, it is even more challenging to verify the reported results. Most anomaly detection methods can run on basic GPUs, or only on CPUs, a significant contrast with AnoLLM. A section discussing inference and training times would help clarify this limitation. I consider this to be the paper’s biggest weakness: its most significant limitation is not addressed at all.

In anomaly detection literature, a clear distinction exists between unsupervised and semi-supervised (or uncontaminated unsupervised) anomaly detection. While we can call them unsupervised methods, since they can be applied in both context, it should be noted that the experiments were conducted in a semi-supervised setting. The distinction lies in whether the training set contains anomalies (unsupervised) or not (semi-supervised). Section 2.1 should be revised to clarify this distinction.

**Minor Comments:**

- In the introduction's first line, "specicious" should be corrected to "specious."
- There is a double colon on line 213 ("equation::").
- Figure 2’s title is missing a space between "yellow" and the parentheses.
- Please use conference or journal citations rather than arXiv versions where possible. Below is a list of those I identified:

  - Liron Bergman and Yedid Hoshen. Classification-based anomaly detection for general data. (ICLR 2020)
  - Vadim Borisov, Kathrin Seßler, Tobias Leemann, Martin Pawelczyk, and Gjergji Kasneci. Language models are realistic tabular data generators. (ICLR 2023)
  - Sungwon Han, Jinsung Yoon, Sercan O Arik, and Tomas Pfister. Large language models can automatically engineer features for few-shot tabular learning. (ICML 2024)
  - Edward J Hu, Yelong Shen, Phillip Wallis, Zeyuan Allen-Zhu, Yuanzhi Li, Shean Wang, Lu Wang, and Weizhu Chen. Lora: Low-rank adaptation of large language models. (ICLR 2022)
  - Nayoung Lee, Kartik Sreenivasan, Jason D Lee, Kangwook Lee, and Dimitris Papailiopoulos. Teaching arithmetic to small transformers. (ICLR 2024)
  - Xuannan Liu, Peipei Li, Huaibo Huang, Zekun Li, Xing Cui, Jiahao Liang, Lixiong Qin, Weihong Deng, and Zhaofeng He. Fakenewsgpt4: Advancing multimodal fake news detection through knowledge-augmented lvlms. (MM2024)
  - Victor Livernoche, Vineet Jain, Yashar Hezaveh, and Siamak Ravanbakhsh. On diffusion modeling for anomaly detection. (ICLR 2024)
  - Ilya Loshchilov and Frank Hutter. Decoupled weight decay regularization. (ICLR 2019)
  - Tomás Mikolov, Kai Chen, Greg Corrado, Jeffrey Dean. Efficient estimation of word representations in vector space. (ICLR Workshop 2013)
  - Hu Wang, Guansong Pang, Chunhua Shen, and Congbo Ma. Unsupervised representation learning by predicting random distances. (AJCAI'20)
  - Jiahuan Yan, Bo Zheng, Hongxia Xu, Yiheng Zhu, Danny Chen, Jimeng Sun, Jian Wu, and Jintai Chen. Making pre-trained language models great on tabular prediction. (ICLR 2024)
  - Tianping Zhang, Shaowen Wang, Shuicheng Yan, Jian Li, and Qian Liu. Generative table pretraining empowers models for tabular prediction. (EMNLP 2023)
  - Bingzhao Zhu, Xingjian Shi, Nick Erickson, Mu Li, George Karypis, and Mahsa Shoaran. Xtab: Cross-table pretraining for tabular transformers. (ICML 2023)
  - Yaqi Zhu, Shaofeng Cai, Fang Deng, and Junran Wu. Do LLMs understand visual anomalies? uncovering LLM capabilities in zero-shot anomaly detection. (MM2024)

**Questions:**

- Did the other baseline also use the ensemble of permutations at inference time?
- What explains the discrepancy in the results of ICL in this paper vs the original paper, which was also tested on OODS?
- How did you choose hyperparameters for the baselines?
- Given the observed trend that larger pretrained models do not seem to benefit AnoLLM, this raises the question: what would happen if we trained a model from scratch using the AnoLLM framework? This feels like a natural question that should have been explored. Did you try this? This might weaken the understanding of the text feature of the model, but it would be interesting to see the impact it has on numerical values.
- What is the total computational cost of the experiments?
- Was experimenting with contaminated training data (the truly unsupervised setting) considered? I ask this because reproducing the paper is not easy, and this is also an important task for anomaly detection.
- What steps are you taking to ensure reproducibility? Will the code be released?
- *See weaknesses*.

---

> ### Author Response · Authors · 2024-11-22
> **Thank you for the constructive feedback.**
>
> Thank you for the constructive feedback. We sincerely appreciate your time in reading the paper and we are grateful for your reviews! Our responses are given below.
>
> **Inconsistency of baseline results with previous work (ICL):** We identified a bug in the DeepOD implementation of ICL. To address this, we adapted the original author’s code and reran the method, updating the results in the current version. Our findings align with your prior observations: ICL performs similarly to KNN. Accordingly, we have revised our claim, stating that AnoLLM performs on par with the best-performing baselines on the ODDS benchmark, rather than outperforming deep learning methods.
>
> **Missing diffusion time estimation (DTE) as baselines:** We have included the results of DTE in the current draft. The updated results show that DTE performs worse than AnoLLM on the mixed-type benchmark but is on par with ICL, KNN, and AnoLLM on the ODDS benchmark.
>
> **Column permutation as ensemble strategies:** We note that column permutation is not inherently applicable to other approaches, as they typically process a single vector as input and do not rely on order dependencies. For these methods, permuting columns is equivalent to permuting dimensions within the vectors and can be interpreted as a naive ensemble strategy.
> Furthermore, we implement ensemble strategies whenever specified in the original papers. Specifically, 5 out of the 11 baselines (Iforest, RCA, SLAD, ICL, and REPEN) already employ ensemble strategies, and their reported results are aggregated across multiple models.
>
> **Other evaluation metrics:** We have included other evaluation metrics such as AUC-PR and F1 scores in the appendix. We observe similar trends as in AUC-ROC.
>
> **Distinction between unsupervised and semi-supervised methods:** Thanks for pointing this out. We have clarified it in Section 2.1. For the contaminated unsupervised setting, we expect naively adapting AnoLLM pipeline to it may not perform well as LLMs might overfit to the contaminated samples. To address this, an outlier-robust variant of AnoLLMs could be explored. For instance, one might use efficient techniques like KNN to filter outliers from the training data prior to training AnoLLMs. Another potential approach is to filter out high-loss training samples during training, similar to the robust collaborative autoencoders (RCA) method. Investigating AnoLLM’s performance in a fully unsupervised setting would be an intriguing direction for future work.
>
> **Hyperparameters selection for baselines:** For each method, we picked the best-performing set of hyperparameters given in their original paper. For others not specified, we use the default hyperparameters as suggested by DeepOD and PyOD toolkit. While it is possible to use a held-out set for hyperparameter tuning in experiments, this approach is impractical in the unsupervised setting. Therefore, to ensure a fair comparison, we adopted the same hyperparameter selection approach as used in ADBench.
>
> Additionally, we would like to also emphasize that we do not manually tune hyperparameters for AnoLLM, as its performance stabilizes once the training loss converges. Thus, users can select the batch size and decide whether to use a LoRA adapter based on GPU memory constraints. Afterward, the learning rate and number of training steps can be chosen to minimize the training error. This process does not rely on labeled data and follows standard practices for fine-tuning LLMs.
>
> **Computation costs of AnoLLMs:**  A runtime comparison experiment is presented in Section E of the Appendix. It is important to note that we selected the smallest LLM backbone, containing only 135M parameters, which allows it to be fine-tuned on a single GPU with 24GB of memory. The use of 7 A100 GPUs was solely to expedite the development process.
> The total computational cost of AnoLLM-135M is approximately 90 GPU-hours on a single RTX-A6000 GPU for the whole experiment. This is also included in Section E.
>
> **Reproducibility and Code release:** We recognize that open-source code plays a crucial role in facilitating academic research and ensuring reproducibility. However, as part of an industry lab, we must comply with company policies. The code will be made available on the company's official GitHub once it receives approval from the legal department.
>
> **Regarding your minor comments:** Thanks for pointing out the typos and citation errors. We have corrected typos and replaced arxiv references with conference or journal citations where available.

---

> > ### Comment · Reviewer_Eu7r · 2024-11-26
> >
> > Thank you for the rebuttal. Most of my concerns have been addressed, and my initial worries about training and inference times are now less significant. While Appendix E provides clarification, I think summarizing this in the main text or adding a clear reference to the appendix would be helpful. I appreciate the reframing of the performance claims, as they are now more accurate.
> >
> > I noted that the table with standard deviations has not been updated in Appendix G, and they are missing for F1 scores and AUC-PR.
> >
> > There is also a question that wasn't answered:
> > > Given the observed trend that larger pretrained models do not seem to benefit AnoLLM, this raises the question: what would happen if we trained a model from scratch using the AnoLLM framework? This feels like a natural question that should have been explored. Did you try this? This might weaken the understanding of the text feature of the model, but it would be interesting to see the impact it has on numerical values.
> >
> > I will review my score after this.

---

> ### Author Response · Authors · 2024-12-01
> **Thank you once again for your thorough review.**
>
> Thank you once again for your thorough review. We are pleased that most of your concerns have been addressed. We hope the following clarifications address your remaining concerns:
>
> **Incorporating references into the main text:** We appreciate your feedback on this matter. We have added a new reference to the Evaluation Protocols paragraph in Section 3. Additionally, we have included an overview of the Appendix in Section A. The tables in Appendix H have also been updated to reflect the standard errors for AUC-ROC, F1, and AUC-PR.
>
> **Performance of AnoLLMs without pretrained weights:** We apologize that we did not have sufficient time to run the experiment in our previous response. Following your suggestion, we evaluated the performance of randomly initialized transformers on the ODDS benchmark, which predominantly comprises 98.5% numerical features. To ensure a fair comparison, we used the same model architecture, SmolLM-135M, with identical hyperparameters. The results are summarized in the table below.
> As shown, AnoLLM with pretrained weights slightly outperforms its randomly initialized counterpart in terms of overall average performance. It achieves better performance on 16 out of 30 datasets and matches performance on 4 datasets. Additionally, a visual inspection of the training curves reveals that AnoLLM with pretrained weights converges approximately twice as fast on most datasets. This faster convergence can be attributed to the pretrained LLM providing a better initialization for fine-tuning.
> In contrast, AnoLLM without pretrained weights not only converges more slowly but is also more susceptible to overfitting, as evidenced by its significantly lower training loss. While overfitting may not be a major concern in uncontaminated, unsupervised settings, it could present challenges in contaminated scenarios, where the model risks memorizing anomalous samples.
> As one of the objectives of this paper is to demonstrate that LLMs can be applied to tabular anomaly detection, an interesting direction for future work would be exploring the trade-off between efficiency and accuracy. Based on this ablation study, for datasets dominated by numerical attributes in uncontaminated and unsupervised settings, AnoLLM without pretrained weights may serve as a more efficient alternative when smaller pretrained models are unavailable.
>
> |                    | SmolLM-135M | AnoLLM without pretrained weights |
> | ------------------ | ------ | ------------------------------ |
> | Annthyroid         | 0.927  | 0.93                           |
> | Arrhythmia         | 0.825  | 0.827                          |
> | BreastW            | 0.992  | 0.993                          |
> | Cardio             | 0.94   | 0.935                          |
> | Ecoli              | 0.777  | 0.778                          |
> | ForestCover        | 0.881  | 0.853                          |
> | Glass              | 0.819  | 0.816                          |
> | Heart              | 0.82   | 0.825                          |
> | Http (KDDCUP99)    | 1      | 1                              |
> | Ionosphere         | 0.909  | 0.89                           |
> | Letter Recognition | 0.967  | 0.907                          |
> | Lymphography       | 0.968  | 0.997                          |
> | Mammography        | 0.915  | 0.878                          |
> | Mulcross           | 1      | 1                              |
> | Musk               | 1      | 1                              |
> | Optdigits          | 0.983  | 0.897                          |
> | Pendigits          | 0.971  | 0.988                          |
> | Pima               | 0.663  | 0.649                          |
> | Satellite          | 0.902  | 0.86                           |
> | Satimage-2         | 1      | 0.998                          |
> | Seismic            | 0.712  | 0.737                          |
> | Shuttle            | 1      | 1                              |
> | Smtp (KDDCUP99)    | 0.927  | 0.926                          |
> | Speech             | 0.47   | 0.459                          |
> | Thyroid            | 0.975  | 0.984                          |
> | Vertebral          | 0.565  | 0.415                          |
> | Vowels             | 0.982  | 0.895                          |
> | WBC                | 0.964  | 0.953                          |
> | Wine               | 0.909  | 0.884                          |
> | Yeast              | 0.744  | 0.749                          |
> | Average            | 0.884  | 0.867                          |

---

> > ### Comment · Reviewer_Eu7r · 2024-12-01
> >
> > This is an interesting finding. The final version of the paper would benefit from discussing it and discussing the impact on datasets with more text features.
> >
> > Regarding references in the text, my main concern was about including references or discussions on training and inference times, which remain absent. This is an important aspect of the method that deserves a brief mention in the main text.
> >
> > As most of my concerns have been addressed, I am increasing my score. However, I strongly encourage the authors to consider the two points mentioned above, especially since there appears to be sufficient space within the page limit.

---

> > > ### Author Response · Authors · 2024-12-01
> > >
> > > We sincerely thank you for raising your scores and offering valuable suggestions to improve the paper draft. If the paper is accepted, we will incorporate the two points into the paper.

---

### Official Review · Reviewer_wUsV · 2024-11-03

**Soundness:** 4
**Presentation:** 4
**Contribution:** 3
**Rating:** 8
**Confidence:** 4

**Summary:**

The paper proposes an innovative use of LLMs, that of detecting anomalies from tabular data. It is well written, and gives good results.

**Strengths:**

S1. The problem statement is innovative.

S2. The paper is quite well written. It is easy to follow and logical.

S3. The results are good.

S4. Instead of simply using large LLMs, small variants are explored, and it is shown that they are no less better.

**Weaknesses:**

W1. The effect of number of decimal digits should have been explored in greater detail.

W2. Similarly, the effect of normalization could have been explored in more detail. Although, the effect of raw numbers is seen, how about simply rounding raw number to x number of decimal digits (and not normalizing) to reduce effect of long decimal numbers, and then using them directly?

W3. What is the effect of not permuting the column names, and having a canonical ordering? Are they not supposed to give even better results?

W4. It will be good to highlight some failure cases, both false positives and false negatives, and try and analyze why that happened.

**Questions:**

W1, W2, W3

---

> ### Author Response · Authors · 2024-11-22
> **Thanks for your encouraging words and constructive comments.**
>
> Thanks for your encouraging words and constructive comments. Your questions are answered below.
>
> **(W1 & W2) Effects of number of decimal numbers:** Designing controlled experiments to analyze the effect of the number of digits on raw numbers presents significant challenges. One key difficulty lies in managing the variation in leading zeros, even when the number of significant digits is carefully controlled. For instance, as observed in the ODDS benchmark, the numerical features can range from 10^5 to 10−7. This wide range makes it impractical to control the number of digits in raw data without applying normalization. Therefore, we use normalization so that most numbers can be represented using 2 or 3 digits. In our early experiments, we also tried 10 and 100 bins for equal-width binning but found no significant differences in the outcomes.
>
> **(W3) Effect of not permuting column names** We conducted an ablation study on the effect of random permutations, detailed in Section D. The results indicate that random permutation is a critical component of AnoLLM, and its absence can lead to a significant decline in performance.
>
> **(W4) Failure cases of AnoLLM:** One failure case of AnoLLM is that the negative log-likelihood assigns equal importance to all features, which can be problematic when certain features are more critical than others. For instance, in the wine dataset, we observed that a single feature, Proline, plays a key role in distinguishing anomalies. AnoLLM performs worse on this dataset because it aggregates anomaly scores across all features, diluting the influence of Proline. In contrast, methods like KNN can better identify the importance of Proline, as its significantly larger values dominate the anomaly scores, leading to more accurate predictions.

---

### Official Review · Reviewer_18uY · 2024-11-04

**Soundness:** 3
**Presentation:** 3
**Contribution:** 2
**Rating:** 6
**Confidence:** 4

**Summary:**

The paper proposes AnoLLM, a large language model (LLM) based framework for unsupervised tabular anomaly detection. It utilizes serialization of tabular data into sentences, and finetunes an LLM with the causal language modeling loss. The model have been applied to numerous datasets and shows competitive performances compared to other baselines.

**Strengths:**

- The paper is generally well-written and easy to follow.
- The paper provides theoretical grounds on their model and shows competitive performances compared to other baselines.
- The paper provides some interesting insights on anomaly detection on tabular data with large-language models.

**Weaknesses:**

- An example for anomaly detection (in Figure 1) would be good for reader's understanding of what is an anomaly detection for tabular data. (Possibly an example explicitly included in the dataset used for the experiments)
- From the results, it is difficult to determine which model performs the best without explicit standard deviation (which is provided in the appendix). It would be better to have some simple plots (e.g., critical difference plots) that incorporates some statistical testing to determine which model is performing better.
- While the authors state there is a similar trends for other metrics, it would be good to see the actual results (at least in the appendix), since there is a high imbalance of normal/anomaly.
- The time comparison between the models and the discussion on the trade-off would be very interesting.
- It is confusing to see "50% of normal samples for training" in the evaluation protocol. Clearer description would help better understanding of the experiments.
- One interesting direction for the future works might be to pretrain a model suited for the anomaly detection. Moreover, more elaboration on impacts for the proposed model would be interesting for the conclusion.

**Questions:**

- What is an example of an anomaly detection in tabular data?
- How does the time comparison look for the baselines?
- What is "50% of normal samples for training"? It seems that the proposed model is unsupervised and 50% is used to train the unsupervised loss, but what does it mean for other models?
- Have the authors considered dealing numerical columns separately? If LLM falls behind in modeling numerical values, it would be interesting to a simple concatenation of numerical values with the output of 'serialization + LLM' and applying the proposed loss.
- Have the authors considered other encoding methods for categorical variables? (e.g., Target Encoder or the package of Skrub)

---

> ### Author Response · Authors · 2024-11-22
> **Thank you for taking the time to read and review our paper!**
>
> Thank you for taking the time to read and review our paper! We are grateful for your feedback. Please see our responses below.
>
> **An example for anomaly detection:** Thank you for your suggestion. We have updated Figure 1, making the anomalous example more clearly visible.
>
> **Statistical testing for clearer comparison:** We provide critical difference plots in Section H of the supplementary material to compare all models on the mixed datasets and the ODDS benchmark. Additionally, we include standard error bars for the ODDS benchmark in Figure 2.
>
> **Results of other performance metrics:** We have included other evaluation metrics such as AUC-PR and F1 scores in the supplementary. We observe the same trends in these metrics. The averaged ranking can also be seen in the critical difference diagrams.
>
> **Tradeoff between runtime and performance:** We provide a compute efficiency analysis in Section F of the Appendix, comparing the runtime of various methods. While AnoLLM is slower due to its large backbone LLM, its performance on datasets with mixed-type data is significantly superior, often outperforming other approaches by a considerable margin. This highlights an interesting trade-off for practitioners to consider: balancing computational efficiency with model efficacy. Future work could explore improving AnoLLM's efficiency through techniques like model distillation or by leveraging it as a feature extractor for classical anomaly detection methods.
>
> **Regarding evaluation protocols:** Our experiments are conducted in an uncontaminated unsupervised setting, where the training set contains only normal samples. Since the datasets lack a predefined train-test split, we randomly partition them to measure performance. Specifically, “50% of normal samples for training” indicates that we randomly select 50% of the normal samples for the training set, while the remaining normal samples, along with all anomalies, are included in the test set. We have clarified it in the evaluation protocol section.
>
> **Other methods for numerical feature modeling:** One of the primary goals of this paper is to demonstrate that LLMs can effectively perform tabular anomaly detection. To achieve this, we use the original pre-processing techniques for numerical features and find that a simple normalization approach performs well. While integrating a separate, specialized numerical encoder (e.g. FT-transformer) is a plausible direction, it would likely demand extensive training data to fine-tune LLMs and enable them to interpret the encoder's outputs. Additionally, adapting the next-token prediction loss to handle continuous outputs, such as numbers, would be necessary. We consider this an intriguing avenue for future research and a potential first step toward developing a foundation model for tabular anomaly detection.
>
> **Other methods for categorical feature modeling:**  Since our task is unsupervised anomaly detection, using a target encoder is not suitable as it requires labeled data. For AnoLLMs, one advantage of leveraging large language models is their ability to interpret raw text directly, so we believe using raw features is the most effective approach. For baseline methods, high-cardinality categorical features can pose challenges for certain approaches. To address this, we group rare categories (those appearing in less than 1% of samples) to reduce the number of classes.
>
> **Pretrained models for anomaly detection:** Thank you for your suggestion. We have incorporated a discussion on the impact of our AnoLLM in the revised conclusion section.

---

> ### Comment · Reviewer_18uY · 2024-11-29
>
> Thank to the authors for making detailed revision on the manuscript.
> The authors have addressed the points that were made and changed the score accordingly.

---

### Author Response · Authors · 2024-11-22

We sincerely thank all reviewers for dedicating their time and effort to evaluating our paper. Your valuable comments and suggestions are greatly appreciated. We have revised the draft accordingly, highlighting the updated text in red, and provided a summary of the changes below:


**Additional evaluation metrics:** We have included other evaluation metrics such as AUC-PR and F1 scores in Section H in the Appendix. We also add critical difference diagrams for statistical ranking comparison (Figure 6 and Figure 7). We observe the same trends in these metrics.

**Modifications to baseline models:** We found that there is a bug in the public implementation of the ICL methods.  To address this, we adapted the original code provided by the authors and re-ran the experiments. The updated results show that ICL performs worse than AnoLLM on the mixed-type benchmark but is on par with KNN and AnoLLM on the ODDS benchmark. These corrected results are reflected in the current version.

**Adding diffusion time estimation (DTE) as baselines:** As pointed out by reviewer 3, we also compare the performance of the DTE.  The updated results show that DTE performs worse than AnoLLM on the mixed-type benchmark but is on par with ICL, KNN, and AnoLLM on the ODDS benchmark.

**Adding runtime comparison:** We provide a compute efficiency analysis in Section F of the Appendix, comparing the runtime of various methods. While AnoLLM is slower due to its large backbone LLM, its performance on datasets with mixed-type data is significantly superior, often outperforming other approaches by a considerable margin. This highlights an interesting trade-off for practitioners to consider: balancing computational efficiency with model efficacy. Future work could explore improving AnoLLM's efficiency through techniques like model distillation or by leveraging it as a feature extractor for classical anomaly detection methods.

**Adding an ablation study - effect of random permutation:** As pointed out by reviewers 2 and 4, we also study the performance of AnoLLM without doing random permutations. The updated results in the Section D show that random permutation does provide significant performance improvements on the mixed typed benchmark.

**Update to the overview figure:** In response to reviewer 1's suggestion, we have added an anomaly example (Figure 1) to better highlight our anomaly detection task.

**Correct typos and arxiv references:** We corrected typos and replaced arxiv references with conference or journal citations where available.

**Code release:** Since we are part of an industry lab, we must adhere to company policies. Therefore, the code will be released on the company's official github once we obtain approval from the legal department.

---

### Author Response · Authors · 2024-11-28

**Additional appendix section:** we have included an overview of the Appendix in Section A.

---

### Meta-Review · Area_Chair_BAXW · 2024-12-20

**Metareview:**

The paper presents a contribution for anomaly detection based on Large Language Models (LLMs) tailored for unsupervised tabular anomaly detection: AnoLLM. A predefined template is used to serialize, i.e., convert tabular data into text for the LLM, along with a preprocessing to mitigate length bias due to the model's autoregressive nature. A negative log-likelihood across different column permutations is used to compute an anomaly score.  A large experimental evaluation is proposed.

Strengths:
-paper well-written,
-novel model based on LLMs,
-solid work,
-competitive performances,
-mitigation of length-bias,
-capacity to deal with tabular data.

Weaknesses:
-some elements need more precisions,
-the impact of normalisation and not permute columns should have been assessed,
-costly method,
-experimental evaluation could be enlarged (e.g. include other strategies like column permutations and other ablation studies),

During the rebuttal, authors have answered to the issues raised by the reviewers, they provided additional experiments including ablation studies and updated their paper with most to the new elements. During the discussion, if it was raised that the explanation about the behavior 7B LLMs was not sufficient for some reviewers the general trend is that the work is interesting and of interest quality. Globally, the answers provided convinced the reviewers and have helped to strengthen the paper.
There is a general consensus for acceptance.

I propose then accept.
I encourage the authors to include the additional elements they have committed to adding.

**Additional Comments On Reviewer Discussion:**

After rebuttal, the reviewers were in general strongly satisfied with the answers.
Reviewer wUsV maintained his strong positive score and Eu7r increased his to score to 8 also.
Reviewer 18uY was satisfied with the answers and increased his score to 6.
Reviewer Sz6K acknowledged that authors have answered to his concerns but had still some reservations and kept his score to 5.

During the discussion, reviewer Sz6K mentioned the explanation on the behavior of 7B LLM models was not convincing but was not opposed to acceptance. Reviewers wUsV and  Eu7r maintained their strong support and reviewer 18uY already indicated his positive feedback.
Acceptance is then naturally proposed.

---

### Decision · Program_Chairs · 2025-01-22

Accept (Poster)